# EEG-Based Multimodal Learning via Hyperbolic Mixture-of-Curvature Experts

**Runhe Zhou**[1][*]  **Shanglin Li**[2][3][4][*]  **Guanxiang Huang**[5]  **Xinliang Zhou**[1]  **Qibin Zhao**[4]  **Motoaki Kawanabe**[3]
**Yi Ding**[1][†]  **Cuntai Guan**[1][†]

## Abstract

Electroencephalography (EEG)-based multimodal learning integrates brain signals with complementary modalities to improve mental state assessment, providing great clinical potential. The effectiveness of such paradigms largely depends on the representation learning on heterogeneous modalities. For EEG-based paradigms, one promising approach is to leverage their hierarchical structures, as recent studies have shown that both EEG and associated modalities (e.g., facial expressions) exhibit hierarchical structures reflecting complex cognitive processes. However, Euclidean embeddings struggle to represent these hierarchical structures due to their flat geometry, while hyperbolic spaces, with their exponential growth property, are naturally suited for them. In this work, we propose EEG-MoCE, a novel hyperbolic mixture-of-curvature experts framework designed for multimodal neurotechnology. EEG-MoCE assigns each modality to an expert in a learnable-curvature hyperbolic space, enabling adaptive modeling of its intrinsic geometry. A curvature-aware fusion strategy then dynamically weights experts, emphasizing modalities with richer hierarchical information. Extensive experiments on benchmark datasets demonstrate that EEG-MoCE achieves state-of-the-art performance, including emotion recognition, sleep staging, and cognitive assessment. Code is available at https://github.com/zhourunhe/EEG-MoCE.

---

[*]Equal contribution  [1]Nanyang Technological University, Singapore [2]BIFOLD, Berlin Institute for the Foundations of Learning and Data, Berlin, Germany [3]ATR, Kyoto, Japan [4]Riken AIP, Tokyo, Japan [5]University of Cambridge, Cambridge, UK. Correspondence to: Yi Ding <ding.yi@ntu.edu.sg>, Cuntai Guan <ctguan@ntu.edu.sg>.

*Proceedings of the 43rd International Conference on Machine Learning*, Seoul, South Korea. PMLR 306, 2026. Copyright 2026 by the author(s).

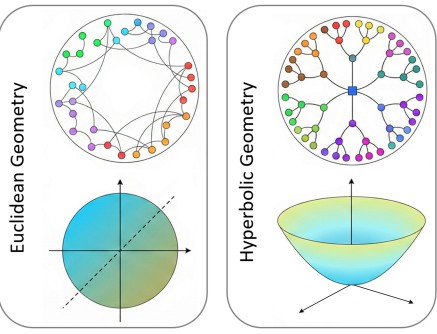

*Figure 1.* **Euclidean vs. hyperbolic geometry for hierarchical data.** Euclidean space is flat and tends to under-represent hierarchical branching; hyperbolic space exhibits exponential volume growth and better preserves tree-like separation. Hyperbolic geometry is informative for multimodal learning, where modalities may differ in how strongly hierarchical their underlying structure is.

## 1. Introduction

Electroencephalography (EEG) records multi-channel electrical activity of the brain (Niedermeyer & Lopes da Silva, 2005) and provides valuable insights into underlying cognitive processes (Bell & Cuevas, 2012). EEG-based neurotechnology aims to extract meaningful patterns to support applications such as sleep stage classification (Aboalayon et al., 2016), cognitive assessment (Shin et al., 2018), and emotion recognition (Suhaimi et al., 2020). However, EEG signals are highly susceptible to external artifacts, and their inherent complexity makes accurately inferring mental states a significant challenge (Lotte et al., 2018; Li et al., 2025).

To overcome these limitations, EEG-based multimodal learning frameworks (Sharma & Meena, 2024), leveraging the strengths of other complementary modalities, have the potential to improve robustness and performance in mental state assessment (Lee et al., 2025). For instance, emotion recognition tasks often benefit from the integration of EEG, facial video, and speech data (Pillalamarri & Shanmugam, 2025). The complementary yet heterogeneous nature stems from their distinct information sources: facial expressions and speech capture observable behavioral cues, while EEG directly measures internal brain activity (Lee et al., 2024).

While integrating these modalities offers a broader perspective, a fundamental challenge remains in effectively repre-

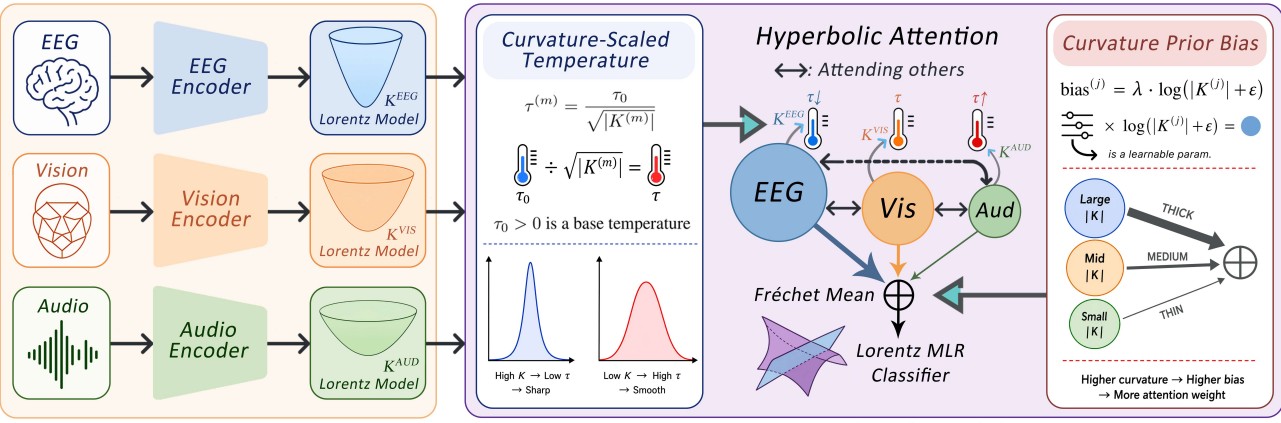

*Figure 2.* **Architecture of EEG-MoCE** on EAV dataset. Other datasets use the same overall architecture, with only modality encoders adapted. (a) **Modality-specific hyperbolic experts**: each modality (e.g., EEG, audio, video) is encoded by an expert that embeds inputs in its own learnable-curvature hyperbolic space. (b) **Curvature-oriented fusion**: expert representations are aggregated by a curvature-aware scheme, combining curvature-scaled temperature and a curvature prior bias, that favors modalities with richer hierarchical structures.

senting heterogeneous modalities (Baltrušaitis et al., 2018). For EEG-based multimodal learning, one promising approach is to leverage their hierarchical structures, as recent studies suggest that the brain's hierarchical cognitive processes can be encoded in EEG (Sun et al., 2023; Turner et al., 2023) and associated modalities (e.g., facial expressions) (Mettes et al., 2024). While Euclidean embeddings currently dominate EEG decoding approaches, they struggle to represent the exponential growth of states in hierarchical structures due to their flat geometry, where distances and areas scale only linearly or quadratically with radius (Peng et al., 2021). In contrast, hyperbolic spaces with negative curvature exhibit exponential growth, making them naturally suited to model such processes. Leveraging this representational advantage, Li et al. (Li et al., 2026) introduced hyperbolic geometry for EEG classification and achieved state-of-the-art (SotA) performance in cross-subject decoding. Beyond EEG, many modalities employed in multimodal neurotechnology, such as facial expression analysis (Mettes et al., 2024) and physiological signal (Búzás et al., 2024), also exhibit inherent hierarchical structures. This shared characteristic motivates the exploration of hyperbolic geometry as a unified embedding framework for EEG-based multimodal learning.

In this work, we introduce EEG-MoCE, a hyperbolic mixture-of-curvature framework for EEG-based multimodal learning (Figure 2), with the following contributions:

- To our knowledge, the first systematic hierarchical analysis and hyperbolic framework for multimodal physiological signals.

- Per-modality experts with learnable curvatures that adapt to intrinsic modality differences.

- Curvature-guided fusion leveraging learned curvatures as proxies for hierarchical structure and modality importance.

- Extensive cross-subject experiments showing strong gains and SotA performance on three public EEG-based multimodal datasets.

Additionally, EEG-MoCE employs recent advances in hyperbolic alignment methods (Li et al., 2026) to mitigate distribution shifts, focusing on the most challenging cross-subject evaluation setting. We organize the paper by quantifying hierarchical structure across modalities, presenting the EEG-MoCE methodology, and reporting cross-subject benchmarks together with analyses and ablations.

## 2. Preliminaries

### 2.1. Related Work

**Cognitive hierarchy.** The human brain exhibits hierarchical organization across multiple cognitive domains. In emotion regulation as an example, initial affective responses originate from subcortical structures, are modulated by limbic circuits, and are ultimately refined by neocortical regions into complex emotional experiences (Panksepp, 2011). Sun et al. (Sun et al., 2023) demonstrated hierarchical emotion ambiguity processing, with distinct EEG patterns revealing the progression from early subcortical responses to later cortical regulation. Hierarchical structure is also implicit in the mechanisms of other modalities (Appendix C.1). Accordingly, hyperbolic geometry supports embeddings of hierarchical data (Krioukov et al., 2010; Peng et al., 2021), motivating a unified geometric view of all modalities in EEG-based multimodal learning.

**EEG-based multimodal learning.** Integrating EEG with complementary modalities has shown promise across emotion recognition (Lee et al., 2024), sleep staging (Jia et al., 2021; Mostafaei et al., 2024), and cognitive assessment (Liu et al., 2025; Shi et al., 2025). However, due to modality characteristics that are fundamentally diverse (Pillalamarri & Shanmugam, 2025), learning unified representations across heterogeneous modalities remains a fundamental challenge (Baltrušaitis et al., 2018). For example, EEG captures internal neural dynamics with inherent variability and noise, while behavioral modalities like facial expressions and speech capture observable and distinct temporal and spatial properties (Lee et al., 2024). To address these challenges, various Euclidean architectures have been proposed, as detailed in Appendix C.3, whereas hyperbolic multimodal learning methods have not yet been explored for EEG-based multimodal learning.

**Hyperbolic and mixed-curvature learning.** Hyperbolic neural networks (Shimizu et al., 2021) have gained significant attention in natural language processing (Ganea et al., 2018) and computer vision (Peng et al., 2021; Mettes et al., 2024) for hierarchical data. In the EEG literature, (Li et al., 2026) demonstrated improved cross-subject generalization by embedding the representation in hyperbolic space. Chang et al. (Chang et al., 2025) performed contrastive pretraining for emotion recognition in hyperbolic space. These methods for neuroscience applications focus on single-modality EEG with fixed curvature. Recent work has shown that for a single textual modality, different components or tasks may exhibit varying degrees of hierarchical structure that are better captured by mixture-of-curvature models (He et al., 2026). Prior work discusses mixed-curvature learning for graph embeddings (Gu et al., 2018) and weighted manifolds for heterogeneous graphs (Nguyen-Van et al., 2023). In this work, we generalize mixture-of-curvature modeling to EEG-based multimodal learning, given that different modalities may exhibit varying degrees of hierarchical structure (Section 4.1). Larger curvature magnitude $|K|$ enables embedding deeper hierarchies with lower distortion without increasing dimensionality (Alvarado et al., 2023), implying that learned curvature can correlate with structural complexity and thus fusion contribution (Section 4.2). We therefore use the learned curvatures to guide cross-modal fusion (Section 3.1).

## 2.2. Hyperbolic Geometry

The hyperbolic space is a Riemannian manifold of constant negative curvature $K < 0$. The Lorentz hyperboloid model is employed in this work owing to its numerical stability during gradient-based optimization (Mishne et al., 2023). The $n$-dimensional Lorentz model represents the upper sheet of a two-sheeted hyperboloid embedded in Minkowski space,

defined as:

$$\mathcal{L}_K^n := \left\{ \mathbf{p} \in \mathbb{R}^{n+1} \,\middle|\, \langle \mathbf{p}, \mathbf{p} \rangle_{\mathcal{L}} = \frac{1}{K}, \; p_t > 0 \right\}, \quad (1)$$

where $\langle \mathbf{p}, \mathbf{q} \rangle_{\mathcal{L}} = \mathbf{p}_s^\top \mathbf{q}_s - p_t q_t$ denotes the Lorentzian inner product. Each point $\mathbf{p} \in \mathcal{L}_K^n$ is decomposed as $\mathbf{p} = [p_t, \mathbf{p}_s^\top]^\top$, where $p_t > 0$ is the time component and $\mathbf{p}_s \in \mathbb{R}^n$ is the space component, following the convention from special relativity (Ratcliffe, 2006). The geodesic distance between two points $\mathbf{p}, \mathbf{q} \in \mathcal{L}_K^n$ is defined as:

$$d_{\mathcal{L}}(\mathbf{p}, \mathbf{q}) = \frac{1}{\sqrt{-K}} \cosh^{-1}\left(K \langle \mathbf{p}, \mathbf{q} \rangle_{\mathcal{L}}\right). \quad (2)$$

Given a collection of points $\{\mathbf{p}_i \in \mathcal{L}_K^n\}_{i=1}^M$ with non-negative weights $\{\eta_i\}_{i=1}^M$ ($\sum_i \eta_i > 0$), the weighted Fréchet mean generalizes the notion of centroid to Riemannian manifolds by minimizing the weighted sum of squared geodesic distances:

$$\boldsymbol{\mu} = \text{wFM}_{\boldsymbol{\eta}}\left(\{\mathbf{p}_i\}_{i=1}^M\right) = \arg \min_{\mathbf{q} \in \mathcal{L}_K^n} \sum_{i=1}^M \eta_i \, d_{\mathcal{L}}^2(\mathbf{q}, \mathbf{p}_i). \quad (3)$$

With uniform weights, this reduces to the standard Fréchet mean. The Fréchet variance $\nu^2$ corresponds to the minimum value achieved at the Fréchet mean.

At each point $\mathbf{p} \in \mathcal{L}_K^n$, the tangent space $T_{\mathbf{p}}\mathcal{L}_K^n$ consists of all vectors $\mathbf{v} \in \mathbb{R}^{n+1}$ satisfying $\langle \mathbf{p}, \mathbf{v} \rangle_{\mathcal{L}} = 0$. The exponential map $\exp_{\mathbf{p}}^K : T_{\mathbf{p}}\mathcal{L}_K^n \to \mathcal{L}_K^n$ projects vectors from the tangent space onto the manifold, while the logarithmic map $\log_{\mathbf{p}}^K : \mathcal{L}_K^n \to T_{\mathbf{p}}\mathcal{L}_K^n$ performs the inverse projection. The parallel transport operation $\text{PT}_{\mathbf{p} \to \mathbf{q}}(\mathbf{v})$ maps tangent vectors from $T_{\mathbf{p}}\mathcal{L}_K^n$ to $T_{\mathbf{q}}\mathcal{L}_K^n$ while preserving geometric properties (see Appendix A for closed-form expressions).

**Hyperbolic neural networks.** Hyperbolic neural networks operate directly on the Lorentz manifold (Bdeir et al., 2024). The Lorentz fully-connected layer transforms features while maintaining the manifold constraint. Given input $\mathbf{p} = [p_t, \mathbf{p}_s^\top]^\top \in \mathcal{L}_K^n$, the transformation follows (Yang et al., 2024):

$$f_{\mathcal{L}}(\mathbf{p}) = \begin{pmatrix} \sqrt{\|\tilde{\mathbf{p}}_s\|^2 - 1/K} \\ \tilde{\mathbf{p}}_s \end{pmatrix}, \quad \text{where } \tilde{\mathbf{p}}_s = \psi(\mathbf{W}\mathbf{p} + \mathbf{b}),$$

$$(4)$$

producing $\tilde{\mathbf{p}} = [\tilde{p}_t, \tilde{\mathbf{p}}_s^\top]^\top \in \mathcal{L}_K^{d'}$, where $d'$ denotes the output dimension, $\mathbf{W} \in \mathbb{R}^{d' \times (n+1)}$ acts on the full Lorentz vector, $\mathbf{b} \in \mathbb{R}^{d'}$, and $\psi$ is an optional activation function. Activation functions such as Lorentz ELU operate on the space components and are combined with the time component to maintain the manifold structure. In hyperbolic attention mechanisms, the standard dot-product similarity is replaced with negative squared geodesic distance to com-

pute attention weights (Yang et al., 2024):

$$\alpha_{ij} = \frac{\exp\left(-d_{\mathcal{L}}^2(\mathbf{q}_i, \mathbf{k}_j)/\tau\right)}{\sum_{j'} \exp\left(-d_{\mathcal{L}}^2(\mathbf{q}_i, \mathbf{k}_{j'})/\tau\right)}, \quad (5)$$

where $\mathbf{q}_i, \mathbf{k}_j \in \mathcal{L}_K^n$ denote query and key vectors, and $\tau > 0$ is a temperature parameter. The weighted Fréchet mean (Equation (3)) is used for aggregating value vectors to maintain geometric consistency. The Lorentz multinomial logistic regression (MLR) classifier extends the Euclidean MLR by measuring distances from data points to decision hyperplanes in hyperbolic space (see Appendix A for details).

**$\delta$-hyperbolicity.** Khrulkov et al. (Khrulkov et al., 2020) introduced $\delta$-hyperbolicity as a metric to assess the tree-like structure inherent in embeddings. This measure determines the minimal value of $\delta$ such that the four-point condition, expressed through the Gromov product, is satisfied (mathematical formulation in Appendix A.5). The Gromov product-based characterization of hyperbolic spaces implies that metric relationships among any four points approximate those in a tree structure, with deviations bounded by an additive constant $\delta$. Smaller values of $\delta \geq 0$ indicate embeddings that are more closely aligned with hyperbolic geometry. To facilitate comparison across modalities, the diameter-normalized $\delta_{\text{rel}} \in [0, 1]$ is computed for each modality (Appendix A.5).

## 3. Methods

### 3.1. Curvature-Oriented Cross-Modal Fusion

Let $\mathcal{M}$ denote the set of modalities and $d$ denote the feature dimension. In our mixture-of-curvature framework, each modality $m \in \mathcal{M}$ is embedded into its own Lorentz manifold $\mathcal{L}_{K^{(m)}}^d$ with learnable curvature $K^{(m)} < 0$. This design is motivated by the observation that mixed-curvature representations can better capture heterogeneous data structures (Gu et al., 2018), and extends naturally to multimodal settings where each modality may possess distinct geometric properties, as we demonstrate empirically in Section 4.1.

**Curvature as information indicator.** The central idea of our fusion strategy is that curvature magnitude serves as a learned geometric indicator of hierarchical complexity. Theoretically, hyperbolic spaces with larger $|K|$ can embed deeper hierarchies with lower distortion without increasing embedding dimensionality (Sala et al., 2018; Alvarado et al., 2023). Since different modalities may exhibit varying degrees of hierarchical structure (as shown in Section 4.1), when curvatures are learned end-to-end, modalities exhibiting richer hierarchical structure will naturally converge to larger $|K|$ values to minimize embedding distortion (Nguyen-Van et al., 2023; Gu et al., 2018). This provides a data-driven signal for modality importance:

larger curvature magnitudes correspond to modalities carrying richer information, which can guide cross-modal fusion.

**Curvature-guided attention mechanism.** Based on this, we design a cross-modal attention mechanism where curvature explicitly guides information aggregation. Our attention mechanism extends Equation (5) using negative squared hyperbolic distance as similarity. Each modality $m$ attends to other modalities with a curvature-scaled temperature:

$$\tau^{(m)} = \frac{\tau_0}{\sqrt{|K^{(m)}|}}, \quad (6)$$

where $\tau_0 > 0$ is a base temperature hyperparameter. Higher $|K|$ yields lower temperature, producing sharper attention distributions. This enables modalities with richer hierarchical structure to be more selective in aggregating cross-modal information.

Furthermore, we incorporate a learnable curvature-based prior into the attention computation to explicitly weight contributions from hierarchically-rich modalities:

$$\tilde{\alpha}_{m \rightarrow j} \propto \exp\left(\frac{-d_{\mathcal{L}}^2(\mathbf{q}^{(m)}, \mathbf{k}^{(j)})}{\tau^{(m)}} + \lambda \cdot \phi(K^{(j)})\right), \quad (7)$$

where $\mathbf{q}^{(m)}, \mathbf{k}^{(j)} \in \mathcal{L}_{K_f}^d$ are query and key vectors after projection to the fusion manifold, $d_{\mathcal{L}}^2$ denotes the squared geodesic distance on the Lorentz model (Equation (2)), $\phi(K) = \log(|K| + \epsilon)$ is a log-curvature transform with small constant $\epsilon > 0$ for numerical stability, and $\lambda > 0$ is a learnable scalar controlling the strength of the curvature prior, constrained to be positive via softplus reparameterization $\lambda = \log(1 + e^{\tilde{\lambda}})$ with unconstrained $\tilde{\lambda} \in \mathbb{R}$. The attended features are aggregated via the weighted Fréchet mean (Equation (3)) on the fusion manifold, preserving hyperbolic geometry throughout the fusion process.

Following (He et al., 2026; Yang et al., 2024), we project each modality into a shared fusion manifold $\mathcal{L}_{K_f}^d$ with curvature-dependent scaling (Equation (8)). The fusion curvature is computed as the mean of modality curvatures: $K_f = \frac{1}{|\mathcal{M}|} \sum_{m \in \mathcal{M}} K^{(m)}$. For a representation $\mathbf{z}^{(m)} \in \mathcal{L}_{K^{(m)}}^d$, the projection proceeds via logarithmic and exponential maps with a curvature-dependent scaling factor:

$$\mathbf{z}_f^{(m)} = \exp_{\mathbf{o}}^{K_f}\left(\sqrt{\frac{K^{(m)}}{K_f}} \cdot \log_{\mathbf{o}}^{K^{(m)}}(\mathbf{z}^{(m)})\right). \quad (8)$$

This is a well-defined map between Lorentz models under which the projected point satisfies the hyperboloid constraint on the shared fusion manifold $\mathcal{L}_{K_f}^d$. The scaling factor $\sqrt{K^{(m)}/K_f}$ preserves the pairwise ordering induced by Lorentz geodesic distances (Yang et al., 2024), and the corresponding reverse projection is likewise well defined (He et al., 2026).

## 3.2. EEG-MoCE Framework

Following the principle of compositional design (Li et al., 2026) that incorporates the advantages of both Euclidean and hyperbolic spaces, we structure EEG-MoCE as a composition of four modules: modality-specific Euclidean encoders $\{e_\theta^{(m)}\}_{m \in \mathcal{M}}$, mixture-of-curvature experts $\{E_\phi^{(m)}\}_{m \in \mathcal{M}}$, a cross-modal fusion module $F_\omega$, and a hyperbolic classifier $g_\psi$. The complete model is parameterized as:

$$h_\Theta = g_\psi \circ F_\omega \circ \left( \bigoplus_{m \in \mathcal{M}} E_\phi^{(m)} \circ e_\theta^{(m)} \right), \qquad (9)$$

where $\Theta = \{\theta, \phi, \omega, \psi, \{K^{(m)}\}_{m \in \mathcal{M}}\}$ includes all learnable parameters. Figure 2 illustrates the overall architecture.

**Modality-specific Euclidean encoders.** Each modality $m$ is first processed by a domain-appropriate Euclidean encoder $e_\theta^{(m)}$. For EEG signals, we adopt EEG-Net (Lawhern et al., 2018) with temporal convolution for frequency-specific filtering, depthwise spatial convolution for electrode-wise patterns, and separable temporal convolution for temporal summarization. For other peripheral physiological signals like EMG and electrooculography (EOG), we use EEGNet-like CNN backbone with similar temporal convolution and spatial convolution for feature extraction. For video (facial expressions), we use a lightweight CNN backbone and apply a temporal transformer for long-range temporal modeling. For audio, we employ a 1D convolutional network operating on mel-spectrograms with a similar temporal transformer architecture as for video. These encoders output feature vectors $\mathbf{x}^{(m)} \in \mathbb{R}^d$ for subsequent hyperbolic processing.

**Mixture-of-curvature experts.** Each modality expert $E_\phi^{(m)}$ operates in its own Lorentz manifold $\mathcal{L}_{K^{(m)}}^d$ with learnable curvature $K^{(m)} < 0$. Unlike prior works using fixed curvatures (Ganea et al., 2018; Chami et al., 2019), learnable curvatures allow the model to adaptively discover the optimal embedding space for each modality (Bdeir et al., 2024; Tan et al., 2024). The expert first projects Euclidean features onto the manifold via the exponential map at the origin:

$$\mathbf{h}^{(m)} = \exp_{\mathbf{o}}^{K^{(m)}}(\mathbf{x}^{(m)}), \qquad (10)$$

where $\mathbf{o} = [\sqrt{-1/K^{(m)}}, \mathbf{0}]^\top$ is the Lorentz origin. The expert then applies hyperbolic batch normalization with moments alignment (Li et al., 2026) for cross-subject generalization, followed by Lorentz activation and pooling (Appendix A), producing the modality-specific hyperbolic representation $\mathbf{z}^{(m)} \in \mathcal{L}_{K^{(m)}}^d$.

**Fusion and classification.** The fusion module $F_\omega$ implements the curvature-oriented cross-modal fusion described in Section 3.1. All modality representations are projected

to the unified fusion manifold $\mathcal{L}_{K_f}^d$ via Equation (8). Multiple curvature-guided cross-attention layers (Equation (7)) are then stacked. The curvature prior bias term $\lambda \cdot \phi(K^{(j)})$ in Equation (7) is applied only in the first layer to keep the modality order information intact before mixing. Each cross-attention layer is followed by hyperbolic layer normalization, and multi-head attention outputs within each layer are aggregated via Fréchet mean (Equation (3)). The final fused representation is obtained by aggregating all modality features via Fréchet mean followed by a hyperbolic linear projection (Equation (4)). Classification is performed using hyperbolic multinomial logistic regression (HMLR) (Ganea et al., 2018; Shimizu et al., 2021), which defines decision boundaries as geodesic hyperplanes on the manifold (Appendix A.3.1).

**Pipeline overview.** From a pipeline perspective, the principal equations of EEG-MoCE define a sequential transformation: Equation (10) first embeds features into per-modality manifolds; Equation (8) then projects them onto a shared fusion manifold to enable cross-modal attention. Equation (6) and Equation (7) jointly parameterize the attention mechanism, using learned curvatures to control the weighting and sharpness of information integration. Following this, Equation (3) aggregates the attended features, and Equation (4) performs the final linear map in hyperbolic space before HMLR (Equation (27)). A detailed mapping of these equations to implementation steps and training updates is provided in Appendix A.4.

## 4. Experiments

EEG-based multimodal technology has great application in real-world scenarios, such as emotion recognition, sleep staging classification, and cognitive assessment. *Emotion recognition* enables affective human-computer interaction and continuous mental health monitoring, supporting applications in personalized therapy (Houssein et al., 2022). *Sleep staging* reduces the burden of manual polysomnographic scoring in clinical diagnosis, enabling large-scale screening of sleep disorders (Siddiqui et al., 2013). *Cognitive assessment* supports objective monitoring of cognitive load and fatigue in educational and occupational settings, facilitating adaptive workload scheduling and personalized training (Wallace et al., 2017).

We evaluate EEG-MoCE using cross-subject evaluation on three public multimodal datasets corresponding to these applications. All datasets provide multimodal recordings for cross-subject evaluation, with preprocessing details provided in Appendix C.2. All baselines are detailed in Appendix C.3.

**EAV** (Lee et al., 2024): Multimodal emotion recognition with EEG, audio, and video from 42 participants across 5

emotion classes (neutral, anger, happiness, sadness, calmness). Task-specific baselines for EAV include: MM-DFN (Hu et al., 2022), GA2MIF (Li et al., 2023), AGF-IB (Shou et al., 2024), CMERC (Tu et al., 2024), Hyper-MML (Kang et al., 2026), HEEGNet (Li et al., 2026).

**ISRUC** (Khalighi et al., 2016): Sleep staging with EEG, EMG, and EOG from 10 subjects, scored into 5 stages (Wake, N1, N2, N3, REM). Our evaluation protocol is equivalent to leave-one-subject-out cross-validation on this dataset. Task-specific baselines for ISRUC include: SS-Net (Jia et al., 2021), CMST (Mostafaei et al., 2024), CFS-Net (Cao et al., 2026), MMNet (Lin et al., 2023), XSleep-Fusion (Hu et al., 2025).

**Cognitive** (Shin et al., 2018): N-back working memory task with EEG, EOG, and near-infrared spectroscopy (NIRS) from 26 participants across 3 difficulty levels (0/2/3-back). Task-specific baselines for Cognitive include: STA-Net (Liu et al., 2025), ST2A (Shi et al., 2025), EFDFNet (Xu et al., 2025), TSMMF (Si et al., 2025), EF-Net (Arif et al., 2024).

We additionally consider several general multimodal frameworks across all tasks: LMF (Liu et al., 2018), CTMWA (Zhang et al., 2024), MMML (Wu et al., 2024). For the cross-validation protocol, we use subject IDs as grouping variables. We apply leave-one-group-out cross-validation when the number of subjects is no greater than 10 (ISRUC), and 10-fold grouped cross-validation otherwise (EAV and Cognitive). In the algorithm level, we further treat individual sessions as distinct domains for adaptation (Li et al., 2026). EEG-MoCE is implemented in PyTorch and trained on NVIDIA RTX 4090 GPUs. Models are trained for 100 epochs using the Adam optimizer for Euclidean parameters and the Riemannian Adam optimizer for hyperbolic parameters, with a learning rate of $10^{-3}$ and early stopping (patience=20).

### 4.1. Hierarchical Structure of Multimodal Data

To further justify hyperbolic embedding, we measure the inherent tree-likeness of each modality using $\delta$-hyperbolicity (Appendix A.5) for all modalities in each dataset following (Khrulkov et al., 2020). Lower $\delta_{\mathrm{rel}} \in [0, 1]$ indicates a stronger hierarchical structure and better suitability for hyperbolic embedding.

Table 1 shows the $\delta_{\mathrm{rel}}$ across modalities for both raw data and encoded features. All modalities exhibit hierarchical structure ($\delta_{\mathrm{rel}} < 0.3$), confirming their suitability for hyperbolic embedding. Furthermore, each modality's $\delta_{\mathrm{rel}}$ exhibits different values from others, demonstrating the rationality of modality-specific learnable curvatures. In particular, EEG overall exhibits small $\delta_{\mathrm{rel}}$ across all datasets, ranging from 0.097 to 0.202, reflecting its hierarchy and justifying its highest learned curvature. Raw features exhibit lower $\delta_{\mathrm{rel}}$

*Table 1.* Hyperbolicity $\delta_{\mathrm{rel}}$ measured for raw data and encoded features following (Khrulkov et al., 2020). Lower values indicate stronger hierarchical tree-like structure and better suitability for hyperbolic embedding. All datasets' modalities show hyperbolic structure with $\delta_{\mathrm{rel}} < 0.3$. Modality-wise differences motivate modality-specific learnable curvature.

| Dataset | Modality | Raw | Encoded |
|---|---|---|---|
| EAV ($n = 42$) | EEG | $0.097 \pm 0.021$ | $0.160 \pm 0.013$ |
| | Audio | $0.217 \pm 0.023$ | $0.293 \pm 0.028$ |
| | Video | $0.280 \pm 0.019$ | $0.278 \pm 0.011$ |
| ISRUC ($n = 10$) | EEG | $0.107 \pm 0.010$ | $0.202 \pm 0.019$ |
| | EMG | $0.072 \pm 0.007$ | $0.202 \pm 0.008$ |
| | EOG | $0.137 \pm 0.013$ | $0.253 \pm 0.028$ |
| Cognitive ($n = 26$) | EEG | $0.125 \pm 0.009$ | $0.183 \pm 0.008$ |
| | EOG | $0.220 \pm 0.013$ | $0.262 \pm 0.011$ |
| | NIRS | $0.299 \pm 0.025$ | $0.292 \pm 0.031$ |

than encoded features, suggesting that Euclidean encoders can disrupt the inherent hierarchical structure and introduce geometric distortion, which motivates our hyperbolic processing after initial encoding.

### 4.2. Learned Curvatures and Modal Contributions

To verify that learned curvatures correlate with modality contributions, we analyze the learned curvature magnitudes $|K|$ for each modality and compute their modal contributions in the fusion process. Modal contribution quantifies the amount of attention each modality receives from others in the cross-attention mechanism. Specifically, for each modality $j$, we compute the sum of attention weights received from all other modalities: $\sum_{m \neq j} \alpha_{m \to j}$, where $\alpha_{m \to j}$ denotes the attention weight from modality $m$ to modality $j$. These values are normalized across all modalities to obtain percentage contributions. We conduct this analysis using a model with learnable curvatures but without curvature-oriented fusion (Section 3.1) to isolate the effect of learned curvatures.

Table 2 presents the results on the EAV dataset. EEG learns the highest curvature magnitude $|K| = 2.34$, reflecting its complex frequency-band hierarchy, followed by Vision ($|K| = 2.29$) and Audio ($|K| = 1.91$). The curvature magnitudes order is consistent with the hierarchical structure of the data revealed by $\delta_{\mathrm{rel}}$. Moreover, results demonstrate that EEG contributes the most (36.0%), consistent with its highest curvature magnitude. The positive correlation between curvature magnitude $|K|$ and contribution confirms that modalities with higher curvature magnitudes receive more attention in the fusion process, validating our curvature-oriented fusion design hypothesis.

Another experimental verification of our curvature-oriented

*Table 2.* Encoded $\delta_{\rm rel}$, learned curvature values $K$ and attention-based modality contributions on EAV dataset, evaluated using a model with learnable curvatures but without curvature-oriented fusion. A negative association between $\delta_{\rm rel}$ and $|K|$ and a positive correlation between $|K|$ and attention contribution support the curvature-oriented fusion hypothesis.

| Modality | Encoded $\delta_{\rm rel}$ | $K$ | Contribution |
|---|---|---|---|
| EEG | 0.160 | -2.34 | 36.0 |
| Vision | 0.278 | -2.29 | 33.6 |
| Audio | 0.293 | -1.91 | 30.5 |

fusion design is the behavior of the parameter curvature prior $\lambda$ in (Equation (7)), which also adapts during training. Table 3 shows that $\lambda$ consistently increases from initialization across all datasets, indicating that the model learns to rely on curvature information for attention weighting.

*Table 3.* Learned curvature prior $\lambda$, initialized at 0.30 on full model with curvature-oriented fusion. Values are mean $\pm$ std across cross-validation folds. $\lambda$ consistently increases during training across datasets, indicating growing reliance on curvature information for attention weighting.

| Dataset | $\lambda_{\rm init}$ | $\lambda_{\rm learned}$ | $\Delta\lambda$ |
|---|---|---|---|
| EAV | 0.300 | $0.464 \pm 0.036$ | +0.164 |
| ISRUC | 0.300 | $0.533 \pm 0.034$ | +0.233 |
| Cognitive | 0.300 | $0.332 \pm 0.009$ | +0.032 |

### 4.3. Main Experiments

Tables 4 to 6 present the comparison results on three datasets: EAV (Lee et al., 2024), ISRUC (Khalighi et al., 2016), and Cognitive (Shin et al., 2018), respectively. The evaluation metrics are balanced accuracy and F1 macro, defined in Appendix B.1.

*Table 4.* Performance Comparison with State-of-the-Art Methods on EAV Emotion Recognition Dataset. We evaluate all methods using balanced accuracy (Acc) and macro-averaged F1-score (F1) under cross-subject 10-fold cross-validation protocol ($n = 42$ subjects).

| Method | Acc (%) | F1 (%) |
|---|---|---|
| MM-DFN | $51.13 \pm 6.81$ | $50.17 \pm 6.95$ |
| CTMWA | $52.86 \pm 5.63$ | $49.92 \pm 6.33$ |
| AGF-IB | $54.59 \pm 12.45$ | $54.20 \pm 15.03$ |
| LMF | $57.29 \pm 12.37$ | $56.80 \pm 14.16$ |
| CMERC | $58.57 \pm 11.95$ | $57.70 \pm 14.30$ |
| MMML | $59.60 \pm 11.83$ | $56.55 \pm 14.31$ |
| Hyper-MML | $60.76 \pm 12.15$ | $57.76 \pm 13.89$ |
| GA2MIF | $61.10 \pm 12.51$ | $58.46 \pm 14.63$ |
| HEEGNet | $61.74 \pm 8.33$ | $60.48 \pm 8.24$ |
| Ours | $\mathbf{75.88} \pm 8.31$ | $\mathbf{75.47} \pm 8.66$ |

*Table 5.* Performance Comparison with State-of-the-Art Methods on ISRUC-S3 Sleep Staging Dataset. We evaluate all methods using balanced accuracy (Acc) and macro-averaged F1-score (F1) under leave-one-subject-out cross-validation protocol ($n = 10$ subjects).

| Method | Acc (%) | F1 (%) |
|---|---|---|
| CMST | $66.68 \pm 8.31$ | $64.06 \pm 10.32$ |
| MMML | $73.33 \pm 3.93$ | $72.80 \pm 4.55$ |
| CFSNet | $74.30 \pm 4.82$ | $72.71 \pm 5.31$ |
| SSNet | $74.56 \pm 4.10$ | $74.13 \pm 4.83$ |
| MMNet | $74.73 \pm 4.02$ | $72.90 \pm 5.45$ |
| LMF | $75.02 \pm 4.54$ | $74.31 \pm 5.63$ |
| CTMWA | $75.11 \pm 3.96$ | $74.20 \pm 4.46$ |
| XSleepFusion | $75.19 \pm 5.20$ | $73.62 \pm 5.80$ |
| Ours | $\mathbf{78.53} \pm 2.95$ | $\mathbf{75.38} \pm 4.05$ |

*Table 6.* Performance Comparison with State-of-the-Art Methods on Cognitive N-back Task. We evaluate all methods using balanced accuracy (Acc) and macro-averaged F1-score (F1) under cross-subject 10-fold cross-validation protocol ($n = 26$ subjects).

| Method | Acc (%) | F1 (%) |
|---|---|---|
| MMML | $45.32 \pm 9.37$ | $42.67 \pm 11.81$ |
| STA-Net | $46.63 \pm 11.67$ | $43.12 \pm 15.32$ |
| EFDFNet | $49.15 \pm 12.85$ | $45.00 \pm 15.13$ |
| CTMWA | $49.86 \pm 11.72$ | $45.71 \pm 12.23$ |
| ST2A | $51.28 \pm 11.74$ | $46.08 \pm 13.17$ |
| LMF | $52.71 \pm 11.77$ | $48.70 \pm 13.04$ |
| TSMMF | $53.84 \pm 13.62$ | $50.06 \pm 16.24$ |
| EF-Net | $54.41 \pm 14.03$ | $51.47 \pm 15.18$ |
| Ours | $\mathbf{62.39} \pm 13.07$ | $\mathbf{59.67} \pm 13.08$ |

The comparison results show that our method achieves significant improvements across all datasets. On the EAV dataset, we outperform all baselines with significance ($p \ll 0.01$) by at least $14.14\%$ in accuracy and $14.99\%$ in F1-score, demonstrating the effectiveness of hyperbolic embedding and curvature-oriented fusion for emotion recognition. On the ISRUC dataset, our method achieves a $3.34\%$ accuracy improvement over XSleepFusion, with notably lower variance, indicating more stable performance across subjects. On the Cognitive dataset, we achieve a $7.98\%$ accuracy improvement over EF-Net, the best baseline, suggesting that our approach is effective for cognitive assessment tasks. The consistent improvements across diverse tasks and modalities validate the generalizability of our framework.

### 4.4. Ablation Study

We conduct comprehensive ablation studies on the EAV dataset to investigate the effectiveness of our proposed components. Appendix B.2 shows the single-modality ablation.

**Euclidean vs Hyperbolic Architecture.** We first examine the impact of hyperbolic geometry in both the encoder and fusion stages. Table 7 compares four architectural variants by independently enabling hyperbolic operations in the encoder and fusion modules.

*Table 7.* Architecture ablation on EAV comparing Euclidean and hyperbolic choices for encoder and fusion. Four variants are shown: neither hyperbolic; hyperbolic fusion only; hyperbolic encoder only; both encoder and fusion. The combination of hyperbolic encoder and hyperbolic fusion produces the best result, indicating complementary benefits.

| Encoder | Fusion | Acc (%) | F1 (%) |
|---------|--------|---------|--------|
| – | – | $60.33 \pm 11.68$ | $57.24 \pm 14.18$ |
| – | ✓ | $61.48 \pm 12.63$ | $58.79 \pm 14.47$ |
| ✓ | – | $74.17 \pm 9.83$ | $73.41 \pm 10.27$ |
| ✓ | ✓ (Ours) | $\mathbf{75.88 \pm 8.31}$ | $\mathbf{75.47 \pm 8.66}$ |

The results demonstrate that hyperbolic geometry in both encoder and fusion stages is essential for optimal performance, with each component contributing complementary benefits. As visualized in Figure 3, the Euclidean encoder and fusion baseline produces severely overlapping emotion clusters, while our hyperbolic encoder and fusion architecture achieves clear class separation with compact, well-defined clusters for each emotion category.

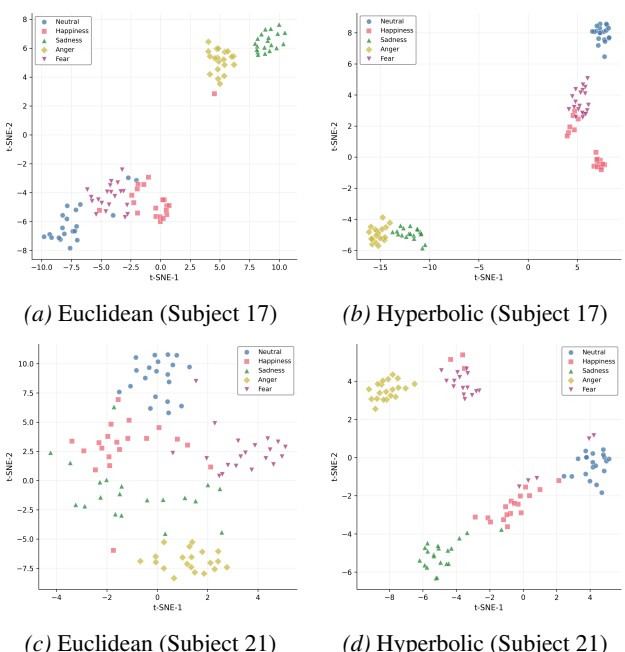

*(a) Euclidean (Subject 17)*    *(b) Hyperbolic (Subject 17)*

*(c) Euclidean (Subject 21)*    *(d) Hyperbolic (Subject 21)*

*Figure 3.* t-SNE of fused features on EAV dataset. Hyperbolic encoder and fusion variant produces more compact and better-separated emotion clusters than the Euclidean baseline variant, illustrating improved class separability by hyperbolic geometry.

**Hyperbolic Component Ablation.** Within the hyperbolic framework, we ablate key design choices: curvature learn-

ing strategy and fusion mechanism. Figure 4 shows the contribution of learnable curvatures and curvature-oriented multimodal fusion (COMF). The fixed curvature baseline is set to $K = -2$ using hyperbolic cross-attention mechanism with COMF. All the variants in fusion comparison utilize the learnable curvatures with initial curvature set to $K = -2$.

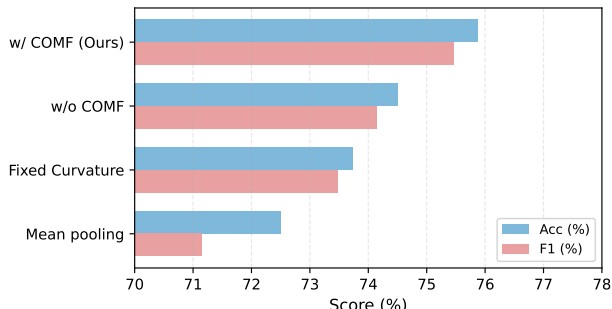

*Figure 4.* Ablation of hyperbolic components on the EAV dataset, focusing on learnable curvature and curvature-oriented multimodal fusion (COMF). All fusion variants use learnable curvatures to isolate the effect of the fusion mechanism. Our proposed method achieves the best performance, demonstrating the complementary benefits of learnable curvatures and COMF.

The results in Figure 4 show that both learnable curvatures and COMF contribute to performance improvements. Compared to fixed curvature, learnable curvatures provide a $2.14\%$ accuracy gain, suggesting that modality-specific geometry optimization is beneficial. COMF contributes a $1.38\%$ accuracy improvement over attention without curvature prior, indicating that incorporating curvature information into attention computation can enhance fusion quality. The combination of learnable curvatures and COMF yields the best performance, suggesting their complementary roles in the fusion process.

### 4.5. Additional Analysis

*Table 8.* Computational cost on EAV measured on the same RTX 4090 GPU. *Train* reports seconds per epoch and *Test* reports milliseconds per trial. EEG-MoCE is slower due to hyperbolic operations, but its absolute inference latency remains low for real-time deployment settings.

| Method | Train sec/epoch | Test msec/trial |
|--------|-----------------|-----------------|
| MMML | 10.86 | 1.17 |
| Hyper-MML | 12.95 | 1.24 |
| GA2MIF | 8.56 | 0.86 |
| Ours | 26.07 | 2.71 |

**Computational cost.** We report training and inference latency on EAV with the same RTX 4090 setup for all methods. Table 8 shows that hyperbolic operations increase computational cost relative to Euclidean baselines, but in-

ference remains lightweight in absolute terms (2.71 ms per 20 s trial), which is still negligible for online BCI scenarios.

**Multi-seed robustness.** To examine initialization sensitivity, we additionally run 5 random seeds and report seed-level statistics while keeping the same data split protocol used in the main experiments. As shown in Table 9, EEG-MoCE consistently outperforms the strongest task-specific baselines on all three datasets, and the seed-level standard deviations are small. This indicates that the reported improvements are not driven by favorable initialization and remain stable across repeated runs.

*Table 9.* Seed-level robustness over 5 random seeds under the same cross-subject protocol as the main experiments. Values are mean $\pm$ std over seeds. Across EAV, ISRUC, and Cognitive, EEG-MoCE consistently exceeds the strongest baseline with low seed-level variance, supporting initialization robustness.

| Dataset | Method | Acc (%) | F1 (%) |
|---|---|---|---|
| EAV | Ours | **75.79** $\pm$ 0.78 | **75.25** $\pm$ 0.76 |
| | HEEGNet | 61.54 $\pm$ 0.76 | 60.37 $\pm$ 0.77 |
| ISRUC | Ours | **78.42** $\pm$ 0.31 | **75.10** $\pm$ 0.29 |
| | XSleepFusion | 75.02 $\pm$ 0.67 | 73.30 $\pm$ 0.71 |
| Cognitive | Ours | **61.86** $\pm$ 1.75 | **59.23** $\pm$ 1.64 |
| | EF-Net | 54.28 $\pm$ 2.41 | 51.36 $\pm$ 2.35 |

## 5. Discussion

In this work, we present EEG-MoCE, the first hyperbolic framework for EEG-based multimodal learning that uses modality-specific learnable curvatures to capture varying hierarchical structure across modalities. Experiments on emotion recognition, sleep staging, and cognitive assessment show state-of-the-art performance. Analysis of learned curvatures demonstrates that they correlate with modality contributions, together with ablation studies validating our geometry-aware design. The consistent gains over prior methods, including hyperbolic baselines, underscore the value of cross-modal synergies in neurotechnology.

While our framework shows promising results, several limitations and future directions warrant discussion. The current approach focuses mainly on hyperbolic geometry and modalities with hierarchical structures; future work could explore other modalities with different structural priors. The computational stability and overhead of hyperbolic operations may suggest potential directions for future research. Although the unified manifold strategy for fusion is empirically effective, richer fusion and transport mechanisms may be explored in future work. Finally, investigating the interpretability of learned representations could provide deeper insights into the underlying cognitive mechanisms.

## Acknowledgements

This work is supported by the Ministry of Education (MOE), Singapore, under its Academic Research Fund Tier 2 (Grant No. MOE-T2EP20124-0001); and is partially supported by the Agency for Science, Technology and Research (A*STAR), Singapore, under its Manufacturing, Trade and Connectivity (MTC) Programmatic Funding Scheme for Scent Digitalization and Computation (SDC) Project (Project No. M23L8b0049). Motoaki Kawanabe is partially supported by the Innovative Science and Technology Initiative for Security Grant Number JPJ004596, ATLA, Japan.

## Impact Statement

This paper presents work whose goal is to advance the field of Machine Learning. There are many potential societal consequences of our work, none of which we feel must be specifically highlighted here.

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

# A. Mathematical Foundations of Hyperbolic Neural Networks

This section provides detailed mathematical foundations for hyperbolic neural network operations used in EEG-MoCE. We organize these operations into three categories: (1) fundamental Riemannian operators that enable mappings between Euclidean and hyperbolic spaces, (2) manifold-preserving neural layers for feature transformation, and (3) task-specific modules for classification and cross-modal fusion. All formulations follow established conventions in hyperbolic deep learning (Ganea et al., 2018; Chami et al., 2019).

## A.1. Fundamental Riemannian Operators

We first establish the core geometric operations that bridge Euclidean preprocessing and hyperbolic representation learning.

**Tangent spaces and their role.** Each point $\mathbf{p}$ on the Lorentz manifold $\mathcal{L}_K^n$ possesses an associated tangent space, a local Euclidean approximation where standard linear algebra applies. Formally, this tangent space is the orthogonal complement under the Lorentzian metric:

$$T_\mathbf{p}\mathcal{L}_K^n = \left\{ \mathbf{v} \in \mathbb{R}^{n+1} \mid \langle \mathbf{p}, \mathbf{v} \rangle_\mathcal{L} = 0 \right\}. \tag{11}$$

**Exponential and logarithmic maps.** These maps provide bidirectional transport between the manifold and tangent spaces (Ratcliffe, 2006). The exponential map $\exp_\mathbf{p}^K : T_\mathbf{p}\mathcal{L}_K^n \to \mathcal{L}_K^n$ projects tangent vectors onto the manifold along geodesics:

$$\exp_\mathbf{p}^K(\mathbf{v}) = \cosh(\alpha)\mathbf{p} + \sinh(\alpha)\frac{\mathbf{v}}{\alpha}, \quad \text{where } \alpha = \sqrt{-K}\|\mathbf{v}\|_\mathcal{L}. \tag{12}$$

Its inverse, the logarithmic map $\log_\mathbf{p}^K : \mathcal{L}_K^n \to T_\mathbf{p}\mathcal{L}_K^n$, retrieves tangent representations:

$$\log_\mathbf{p}^K(\mathbf{q}) = \frac{\cosh^{-1}(\beta)}{\sqrt{\beta^2 - 1}} \cdot (\mathbf{q} - \beta\mathbf{p}), \quad \text{where } \beta = K\langle \mathbf{p}, \mathbf{q} \rangle_\mathcal{L}. \tag{13}$$

For projecting Euclidean features $\mathbf{x} \in \mathbb{R}^n$ onto $\mathcal{L}_K^n$, we use the distinguished origin $\mathbf{o} = [\sqrt{-1/K}, \mathbf{0}^\top]^\top$, where the exponential map simplifies to:

$$\exp_\mathbf{o}^K(\mathbf{x}) = \left( \frac{\cosh(\sqrt{-K}\|\mathbf{x}\|)}{\sqrt{-K}}, \frac{\sinh(\sqrt{-K}\|\mathbf{x}\|)}{\sqrt{-K}\|\mathbf{x}\|} \cdot \mathbf{x} \right)^\top. \tag{14}$$

**Parallel transport.** When displacing tangent vectors between different base points, parallel transport preserves geometric relationships. For $\mathbf{v} \in T_\mathbf{p}\mathcal{L}_K^n$ transported to $T_\mathbf{q}\mathcal{L}_K^n$ (Ganea et al., 2018):

$$\text{PT}_{\mathbf{p}\to\mathbf{q}}(\mathbf{v}) = \mathbf{v} - \frac{K\langle \mathbf{q}, \mathbf{v} \rangle_\mathcal{L}}{1 + K\langle \mathbf{p}, \mathbf{q} \rangle_\mathcal{L}}(\mathbf{p} + \mathbf{q}). \tag{15}$$

## A.2. Manifold-Preserving Neural Layers

Standard neural network operations must be adapted to respect manifold constraints. We describe the key modifications organized by their functional role.

### A.2.1. Nonlinear Activation Functions

Applying nonlinearities directly to Lorentz vectors would violate manifold membership. Following the principle of operating on unconstrained coordinates (Shimizu et al., 2021; Bdeir et al., 2024), we apply activations exclusively to the spatial (spacelike) components $\mathbf{p}_s \in \mathbb{R}^n$, then reconstruct the temporal coordinate to restore the Lorentz constraint $\langle \mathbf{p}, \mathbf{p} \rangle_\mathcal{L} = 1/K$:

$$\sigma_\mathcal{L}(\mathbf{p}) = \begin{pmatrix} \sqrt{\|\sigma(\mathbf{p}_s)\|^2 - 1/K} \\ \sigma(\mathbf{p}_s) \end{pmatrix}, \tag{16}$$

where $\sigma$ denotes any standard activation (e.g., ELU, ReLU). This design ensures the output remains on the hyperboloid regardless of the activation choice.

A.2.2. FEATURE AGGREGATION AND POOLING

Arithmetic averaging is ill-defined on curved manifolds. Instead, spatial pooling computes the Fréchet mean (Fréchet, 1948) over features within each receptive field. Given features $\{\mathbf{p}_i\}_{i=1}^{M}$ in a pooling window, the aggregated output minimizes total squared geodesic distance (Equation (3) in main text).

A.2.3. HYPERBOLIC BATCH NORMALIZATION

Batch normalization (Ioffe & Szegedy, 2015) is essential for stable training but requires adaptation to respect manifold geometry. Following the gyrovector formalism (Chen et al., 2025b; Li et al., 2026), we define hyperbolic batch normalization (HBN) using Lorentz gyrovector operations that generalize Euclidean centering and scaling.

**Gyrovector operations.** The Lorentz model admits a gyrovector structure where standard vector operations are replaced by their hyperbolic analogues (Ungar, 2008). Specifically, gyroaddition $\oplus$, gyroinverse $\ominus$, and gyroscalar multiplication $\odot$ are defined via exponential and logarithmic maps:

$$\mathbf{p} \oplus \mathbf{q} = \exp_{\mathbf{o}}\left(\mathrm{PT}_{\mathbf{o} \to \mathbf{p}}(\log_{\mathbf{o}}(\mathbf{q}))\right), \tag{17}$$

$$\ominus \mathbf{p} = [p_t, -\mathbf{p}_s]^{\top}, \tag{18}$$

$$t \odot \mathbf{p} = \exp_{\mathbf{o}}(t \cdot \log_{\mathbf{o}}(\mathbf{p})), \tag{19}$$

where PT denotes parallel transport (Appendix A.1).

**Batch normalization in hyperbolic space.** Given a batch of activations $\{\mathbf{p}_i \in \mathcal{L}_K^n\}_{i=1}^{M}$, HBN computes the Fréchet mean $\boldsymbol{\mu}$ and Fréchet variance $\nu^2 = \frac{1}{M}\sum_{i=1}^{M} d_{\mathcal{L}}^2(\boldsymbol{\mu}, \mathbf{p}_i)$ as batch statistics. The normalization then centers and scales each point using gyrovector operations:

$$\mathrm{HBN}(\mathbf{p}_i) = \frac{\gamma}{\sqrt{\nu^2 + \epsilon}} \odot \left(\ominus \boldsymbol{\mu} \oplus \mathbf{p}_i\right), \quad \forall i \le M, \tag{20}$$

where $\gamma \in \mathbb{R}$ is a learnable scaling parameter and $\epsilon > 0$ ensures numerical stability. The centering operation $\ominus \boldsymbol{\mu} \oplus \mathbf{p}_i$ translates each point by the inverse of the batch mean, analogous to subtracting the mean in Euclidean BN. The scaling operation adjusts the dispersion via gyroscalar multiplication.

**Domain-specific momentum estimation.** For cross-subject generalization in EEG, we adopt domain-specific momentum batch normalization (Kobler et al., 2022; Li et al., 2026). Each domain $d \in \mathcal{D}$ maintains separate running estimates $(\tilde{\boldsymbol{\mu}}^{(d)}, \tilde{\nu}^{2(d)})$ updated via geodesic interpolation:

$$\tilde{\boldsymbol{\mu}}_k^{(d)} = \mathrm{Geodesic}(\tilde{\boldsymbol{\mu}}_{k-1}^{(d)}, \boldsymbol{\mu}_k^{(d)}; \eta), \tag{21}$$

$$\tilde{\nu}_k^{2(d)} = (1-\eta)\tilde{\nu}_{k-1}^{2(d)} + \eta \cdot \nu_k^{2(d)}, \tag{22}$$

where $\mathrm{Geodesic}(\mathbf{p}, \mathbf{q}; t)$ returns the point at fraction $t \in [0,1]$ along the geodesic from $\mathbf{p}$ to $\mathbf{q}$, and $\eta$ is a momentum parameter following an exponential decay schedule during training. At test time, a fixed momentum $\eta_{\text{test}}$ adapts to unseen target domains in a source-free manner.

**Curvature-adaptive running statistics.** When curvature $K$ is learnable, running statistics must be transformed to preserve geometric relationships. For the running mean $\tilde{\boldsymbol{\mu}}$, we apply log-scale-exp projection:

$$\tilde{\boldsymbol{\mu}}_{\text{new}} = \exp_{\mathbf{o}}^{K_{\text{new}}}\left(\sqrt{\frac{K_{\text{old}}}{K_{\text{new}}}} \cdot \log_{\mathbf{o}}^{K_{\text{old}}}(\tilde{\boldsymbol{\mu}}_{\text{old}})\right). \tag{23}$$

For variance, since $\nu^2 \propto 1/|K|$ (geodesic distance scales with $1/\sqrt{|K|}$), we apply:

$$\tilde{\nu}_{\text{new}}^2 = \tilde{\nu}_{\text{old}}^2 \cdot \frac{K_{\text{old}}}{K_{\text{new}}}. \tag{24}$$

These transformations ensure that normalized features maintain consistent geometric properties as the embedding space curvature evolves during training.

### A.2.4. Dimension Transformation and Feature Projection

Linear transformations in hyperbolic space follow the Hyperbolic Transformation with Curvatures (HTC) framework (Yang et al., 2024). For input $\mathbf{p} = [p_t, \mathbf{p}_s^\top]^\top \in \mathcal{L}_K^n$ with parameters $\mathbf{W} \in \mathbb{R}^{d' \times (n+1)}$ and $\mathbf{b} \in \mathbb{R}^{d'}$:

$$f_{\mathcal{L}}(\mathbf{p}) = \begin{pmatrix} \sqrt{\|\tilde{\mathbf{p}}_s\|^2 - 1/K} \\ \tilde{\mathbf{p}}_s \end{pmatrix}, \quad \text{where } \tilde{\mathbf{p}}_s = \psi(\mathbf{W}\mathbf{p} + \mathbf{b}), \tag{25}$$

with optional activation $\psi$, producing output $\tilde{\mathbf{p}} \in \mathcal{L}_K^{d'}$ where $d'$ denotes the output dimension. This transformation applies to the full Lorentz vector, enabling both Lorentz rotations and boosts (Chen et al., 2022). Curvature changes in our framework are handled separately via log-exp map projection (see Equation (8) in the main text).

**Hyperbolic concatenation.** For spatially-structured inputs, we extend this to convolutions by: (1) gathering features within each receptive field, (2) performing hyperbolic concatenation of their spatial components while preserving manifold structure, and (3) applying the linear transformation. Specifically, for features $\{\mathbf{p}_{h,w}\}$ in a window, the output is $f_{\mathcal{L}}(\text{HCat}(\{\mathbf{p}_{h,w}\}))$, where hyperbolic concatenation (Lorentz direct concatenation (Qu & Zou, 2022)) combines $N$ vectors as:

$$\text{HCat}(\{\mathbf{p}_i\}_{i=1}^N) = \begin{pmatrix} \sqrt{\sum_{i=1}^N p_{it}^2 + (N-1)/K} \\ \mathbf{p}_{1s}^\top, \dots, \mathbf{p}_{Ns}^\top \end{pmatrix}^\top \in \mathcal{L}_K^{nN}. \tag{26}$$

## A.3. Classification and Cross-Modal Fusion

### A.3.1. Hyperbolic Classification via Geodesic Hyperplanes

Classification in hyperbolic space replaces Euclidean linear boundaries with geodesic hyperplanes (Ganea et al., 2018; Shimizu et al., 2021). Each class $c \in \{1, \dots, C\}$ is parameterized by a scalar $a_c \in \mathbb{R}$ controlling hyperplane position and a direction vector $\mathbf{z}_c \in \mathbb{R}^n$. The logit for class $c$ measures the signed distance from input $\mathbf{p} \in \mathcal{L}_K^n$ to the corresponding hyperplane:

$$\ell_c(\mathbf{p}) = \frac{\text{sign}(\alpha_c) \cdot \beta_c}{\sqrt{-K}} \sinh^{-1}\left(\frac{\sqrt{-K}\,\alpha_c}{\beta_c}\right), \tag{27}$$

with intermediate quantities:

$$\alpha_c = \cosh(\sqrt{-K}a_c)\langle \mathbf{z}_c, \mathbf{p}_s\rangle - \sinh(\sqrt{-K}a_c)\|\mathbf{z}_c\| \cdot p_t, \tag{28}$$

$$\beta_c = \sqrt{\|\cosh(\sqrt{-K}a_c)\mathbf{z}_c\|^2 - \left(\sinh(\sqrt{-K}a_c)\|\mathbf{z}_c\|\right)^2}. \tag{29}$$

Points on the positive side of the hyperplane yield positive logits, enabling standard softmax classification.

### A.3.2. Hyperbolic Cross-Modal Attention

Our cross-attention mechanism enables information exchange between modalities while respecting hyperbolic geometry. Unlike self-attention, each modality $m$ attends only to *other* modalities $\mathcal{M} \setminus \{m\}$, ensuring pure cross-modal information flow.

**Attention computation.** Our attention mechanism extends Hypformer's approach (Yang et al., 2024) by using explicit hyperbolic distance computation with curvature-adaptive temperature and curvature-based priors, while maintaining the core idea of using negative squared hyperbolic distance as similarity. After projecting all modalities to the unified fusion manifold $\mathcal{L}_{K_f}^d$, we compute attention weights using negative squared geodesic distance as similarity (Gulcehre et al., 2019; Yang et al., 2024):

$$\alpha_{m \to j} = \frac{\exp\left(-d_{\mathcal{L}}^2(\mathbf{q}^{(m)}, \mathbf{k}^{(j)})/\tau^{(m)}\right)}{\sum_{j' \in \mathcal{M} \setminus \{m\}} \exp\left(-d_{\mathcal{L}}^2(\mathbf{q}^{(m)}, \mathbf{k}^{(j')})/\tau^{(m)}\right)}, \tag{30}$$

where query $\mathbf{q}^{(m)}$ and keys $\mathbf{k}^{(j)}$ are obtained via hyperbolic linear projections (Equation (25)), and $\tau^{(m)} > 0$ is the temperature. Following prior hyperbolic attention formulations, similarity is defined by negative squared geodesic distance, i.e., $\exp(-d_{\mathcal{L}}^2/\tau)$, where $d_{\mathcal{L}}$ is given in Equation (2).

---

**Algorithm 1** EEG-MoCE pipeline

---

**Require:** Minibatch inputs; $\mathbf{y}$ if training; $\{K^{(m)}\}, \tau_0, L$.
**Ensure:** $\hat{\mathbf{y}}$; updated $\Theta$ if training.
 1: **for** each modality $m$ **do**
 2:     $\mathbf{x}^{(m)} \leftarrow e_\theta^{(m)}(\cdot)$ {Euclidean encoding (Sec. 3.2)}
 3:     $\mathbf{h}^{(m)}$ from Equation (10) {lift to per-modality $\mathcal{L}_{K^{(m)}}^d$}
 4:     Equation (20) {BN}; Equation (16) {activation}; Equation (26) {concatenation} $\rightarrow \mathbf{z}^{(m)}$
 5: **end for**
 6: $K_f \leftarrow |\mathcal{M}|^{-1} \sum_m K^{(m)}$
 7: **for** each $m$ **do**
 8:     $\mathbf{z}_f^{(m)}$ via Equation (8) {align all modalities to $\mathcal{L}_{K_f}^d$}
 9: **end for**
10: **for** $\ell = 1, \ldots, L$ **do**
11:     Equation (6) {curvature-scaled temperature}; Equation (7) with $d_\mathcal{L}$ from Equation (2) {hyperbolic attention}; add $\lambda\phi(K^{(j)})$ term of Equation (7) iff $\ell=1$
12:     merge with Equation (3) {Fréchet mean}
13: **end for**
14: Equation (4) {linear}; $\hat{\mathbf{y}} \leftarrow$ Equation (27) {MLR}
15: **if** training **then**
16:     $\mathcal{L} \leftarrow \text{CE}(\hat{\mathbf{y}}, \mathbf{y})$; update $\Theta$ {RSGD (Bonnabel, 2013) where needed}
17: **end if**

---

**Multi-head aggregation.** Each attention head produces an output via weighted Fréchet mean over attended values. Outputs across $H$ heads are then combined via another Fréchet mean, followed by hyperbolic layer normalization (Yang et al., 2024):

$$\mathbf{z}_{\text{cross}}^{(m)} = \text{LN}_\mathcal{L}\left(\text{wFM}\left(\left\{\text{Head}_h^{(m)}\right\}_{h=1}^H\right)\right). \tag{31}$$

Layer normalization applies standard normalization to spatial coordinates, then reconstructs the temporal component (analogous to Appendix A.2.1).

## A.4. EEG-MoCE Algorithm Pipeline

For every expression referenced in Algorithm 1, we state its functional role, linking mathematical notation to implementation logic. The scalar fusion curvature $K_f = |\mathcal{M}|^{-1} \sum_{m \in \mathcal{M}} K^{(m)}$ (Sec. 3.1) fixes the Lorentz scale of $\mathcal{L}_{K_f}^d$ before Equation (8); it is not assigned a separate equation number. Training uses cross-entropy and parameter updates (including Riemannian steps where needed) without additional displayed objects beyond Algorithm 1.

**Modality expert.** Equation (10) lifts $e_\theta^{(m)}(\cdot)$ onto the per-modality Lorentz manifold $\mathcal{L}_{K^{(m)}}^d$. Equation (20) performs gyrovector batch normalization on $\mathcal{L}_{K^{(m)}}^d$. Equation (16) applies $\sigma$ to the spacelike components and reconstructs the time coordinate so the point remains on the hyperboloid. Equation (26) implements the Lorentz direct concatenation.

**Cross-modal fusion.** Equation (8) maps each expert output from $\mathcal{L}_{K^{(m)}}^d$ to the shared $\mathcal{L}_{K_f}^d$ so that the interaction between queries and keys is well defined on a unified Lorentz manifold. Equation (6) sets the per-modality softmax temperature $\tau^{(m)}$. Equation (2) defines the geodesic distance $d_\mathcal{L}$; Equation (7) builds logits from squared hyperbolic distance $d_\mathcal{L}^2/\tau^{(m)}$ and adds the curvature prior $\lambda\phi(K^{(j)})$. Equation (3) fuses attended values inside each fusion layer and performs the final modality-level pool on $\mathcal{L}_{K_f}^d$.

**Classification head.** Equation (4) applies the Lorentz fully connected map to the fused embedding on $\mathcal{L}_{K_f}^d$. Equation (27) maps that representation to class logits via the hyperbolic multinomial logistic regression described in Appendix A.3.1.

## A.5. Quantifying Hierarchical Structure

To validate that target modalities benefit from hyperbolic embedding, we measure their intrinsic tree-likeness via Gromov's $\delta$-hyperbolicity (Gromov, 1987).

**Gromov product.** For a metric space $(X, d)$, the Gromov product quantifies how closely three points approximate a tripod configuration:

$$(\mathbf{y}, \mathbf{z})_{\mathbf{x}} = \frac{1}{2} \left( d(\mathbf{x}, \mathbf{y}) + d(\mathbf{x}, \mathbf{z}) - d(\mathbf{y}, \mathbf{z}) \right). \tag{32}$$

Geometrically, larger values indicate $\mathbf{x}$ lies further from the geodesic between $\mathbf{y}$ and $\mathbf{z}$.

**Four-point condition.** A space is $\delta$-hyperbolic if all quadruples satisfy:

$$(\mathbf{x}, \mathbf{z})_{\mathbf{w}} \geq \min \left\{ (\mathbf{x}, \mathbf{y})_{\mathbf{w}}, (\mathbf{y}, \mathbf{z})_{\mathbf{w}} \right\} - \delta. \tag{33}$$

Trees satisfy this exactly with $\delta = 0$; the parameter $\delta$ measures deviation from perfect tree geometry.

**Scale-invariant metric.** For cross-dataset comparison, we normalize by diameter (Khrulkov et al., 2020):

$$\delta_{\text{rel}}(X) = \frac{2\delta(X)}{\text{diam}(X)} \in [0, 1]. \tag{34}$$

Values near zero indicate strong hierarchical structure amenable to low-distortion hyperbolic embedding.

# B. Additional Experimental Details

## B.1. Definition of Evaluation Metrics

The evaluation metrics are balanced accuracy and F1 macro, defined as:

$$\text{Balanced Accuracy} = \frac{1}{C} \sum_{i=1}^{C} \frac{TP_i}{TP_i + FN_i} \tag{35}$$

$$\text{F1 Macro} = \frac{1}{C} \sum_{i=1}^{C} \frac{2 \cdot TP_i}{2 \cdot TP_i + FP_i + FN_i} \tag{36}$$

where $C$ is the number of classes, $TP_i$ is the true positive count for class $i$, $FP_i$ is the false positive count for class $i$, and $FN_i$ is the false negative count for class $i$.

## B.2. Single-Modality Ablation

We evaluate the complementary relationship of our multimodal framework by training with single modalities. Table 10 shows that multimodal fusion significantly outperforms any single modality. EEG achieves the best single-modality performance (62.74%), consistent with its highest curvature magnitude and largest fusion contribution, suggesting that it carries relatively more discriminative hierarchical information. Multimodal fusion (75.88%) exceeds the best single modality by 13.14%, indicating complementary relationships between modalities. The improvement over individual modality performances suggests that our fusion mechanism may capture cross-modal synergies beyond simple aggregation of independent signals.

*Table 10.* Ablation study on single-modality performance on EAV dataset.

| Modality | Acc (%) | F1 (%) |
|---|---|---|
| Video only | $53.75 \pm 10.74$ | $53.46 \pm 11.02$ |
| Audio only | $60.52 \pm 8.14$ | $60.23 \pm 8.42$ |
| EEG only | $62.74 \pm 9.04$ | $62.13 \pm 9.42$ |
| All modalities | $\mathbf{75.88} \pm 8.31$ | $\mathbf{75.47} \pm 8.66$ |

## B.3. Fusion Curvature Design

We compare the default mean fusion curvature with a learnable fusion curvature. Table 11 shows that the simple mean is consistently comparable or slightly better across all datasets in both accuracy and F1. These results support using mean fusion curvature as a stable and effective default without introducing additional optimization variables.

*Table 11.* Ablation of fusion-curvature parameterization in the shared manifold. *Mean* uses the arithmetic mean of modality curvatures, while *Learnable* treats the fusion curvature as an extra trainable parameter. Mean curvature performs comparably or slightly better across all datasets, supporting the stable and effective design choice.

| Dataset | Fusion | Acc (%) | F1 (%) |
|---|---|---|---|
| EAV | Mean | $\mathbf{75.88} \pm 8.31$ | $\mathbf{75.47} \pm 8.66$ |
| | Learnable | $75.55 \pm 8.57$ | $75.10 \pm 9.13$ |
| ISRUC | Mean | $\mathbf{78.53} \pm 2.95$ | $\mathbf{75.38} \pm 4.05$ |
| | Learnable | $78.08 \pm 3.01$ | $74.97 \pm 3.96$ |
| Cognitive | Mean | $\mathbf{62.39} \pm 13.07$ | $\mathbf{59.67} \pm 13.08$ |
| | Learnable | $60.96 \pm 13.48$ | $57.68 \pm 13.61$ |

To further examine the mean-curvature design, we compare it with an optimization-based fusion curvature computed from pretrained per-modality curvatures. The optimization-based reference follows the perspective introduced in concurrent work on modality alignment across hyperbolic manifolds (Wei et al., 2026). Table 12 shows that the two estimates are statistically

close on all datasets. We perform paired two one-sided tests (TOST) on fold-wise paired means with equivalence margin $\pm 0.005$. The test supports that the mean curvature is an stable approximation for practical use.

*Table 12.* Mean fusion curvature vs. optimization-based fusion curvature (Optimized $K$) derived using the objective in (Wei et al., 2026). Values are mean $\pm$ std across folds. The close agreement across datasets supports the practical use of mean fusion curvature.

| Dataset | Mean $K$ | Optimized $K$ |
|---|---|---|
| EAV | $-2.2544 \pm 0.1020$ | $-2.2509 \pm 0.1013$ |
| ISRUC | $-1.3511 \pm 0.0777$ | $-1.3547 \pm 0.0760$ |
| Cognitive | $-1.9834 \pm 0.0630$ | $-1.9821 \pm 0.0623$ |

## B.4. Additional Experimental Results on Cognitive Dataset

We report additional results on the Word Generation task of the Cognitive dataset. In this task, participants were instructed to think of words beginning with the given letter as fast as possible without repeating the same word; in Baseline, they relaxed and gazed at the fixation cross for low cognitive load. Table 13 compares our method with state-of-the-art approaches under the same cross-subject evaluation protocol.

*Table 13.* Performance comparison with state-of-the-art methods on the Cognitive dataset, Word Generation task ($n = 26$). We report balanced accuracy (Acc) and macro-averaged F1 (F1) under cross-subject 10-fold cross-validation.

| Method | Acc (%) | F1 (%) |
|---|---|---|
| MMML (Wu et al., 2024) | $56.35 \pm 8.22$ | $51.76 \pm 12.07$ |
| EFDFNet (Xu et al., 2025) | $58.85 \pm 7.63$ | $53.83 \pm 11.13$ |
| LMF (Liu et al., 2018) | $59.74 \pm 6.18$ | $59.16 \pm 6.38$ |
| CTMWA (Zhang et al., 2024) | $59.87 \pm 8.15$ | $58.68 \pm 8.63$ |
| STA-Net (Liu et al., 2025) | $62.50 \pm 8.72$ | $58.08 \pm 12.65$ |
| TSMMF (Si et al., 2025) | $63.82 \pm 9.47$ | $62.87 \pm 10.36$ |
| EF-Net (Arif et al., 2024) | $64.53 \pm 11.08$ | $63.43 \pm 12.33$ |
| ST2A (Shi et al., 2025) | $67.27 \pm 10.45$ | $66.64 \pm 12.51$ |
| Ours | $\mathbf{73.65} \pm 6.93$ | $\mathbf{73.36} \pm 7.09$ |

## C. Supplementary Related Work

### C.1. Hierarchical Structures

In this paragraph, we provide additional evidence for the claim that the modalities used in our experiments also exhibit hierarchical structure. In visual processing, elementary features are first extracted by lower cortical regions, then integrated into increasingly complex representations by higher cortical areas (Hochstein & Ahissar, 2002). For scalp EEG, Collins et al. (Collins et al., 2018) used different stimulus frequencies to reveal distinct neural processes at different levels of the visual hierarchy. Chen et al. (Chen et al., 2025a) demonstrated that facial expressions exhibit rich hierarchical relationships, where basic expressions form the foundation for more complex compound expressions. For physiological signals, Búzás et al. (Búzás et al., 2024) revealed hierarchical levels in the temporal organization of human physical activity, where activity bursts at the minute scale are organized into superstructures at longer scales (Lopez et al., 2024). Together with our empirical results on hierarchical structure, this evidence suggests that hyperbolic embeddings are a promising framework for multimodal learning.

### C.2. Dataset Details

We provide detailed descriptions and preprocessing procedures for the three datasets used in our experiments.

**EAV** (Lee et al., 2024) is a multimodal emotion recognition benchmark in conversational contexts, comprising 30-channel EEG, audio recordings of speech, and video recordings of facial expressions from 42 participants. The experiment elicited five emotions (neutral, anger, happiness, sadness, and calmness) through cue-based conversation scenarios, with each participant contributing 100 trials with simultaneous recording of all three modalities. Following the official procedure, we preprocess the dataset by downsampling the EEG signal to 100 Hz and bandpass filtering at 0.5–45 Hz. Audio is resampled to 16 kHz and converted to mel-spectrogram, while visual features are extracted using OpenFace (Baltrušaitis et al., 2016) at 5 fps.

**ISRUC** (Khalighi et al., 2016) is a sleep dataset containing polysomnographic (PSG) recordings, including EEG, EMG, and EOG from 10 healthy subjects. Each recording was scored by two human experts according to standard sleep staging criteria (Wake, N1, N2, N3, and REM). All PSG signals are resampled to 100 Hz and bandpass filtered (EEG: 0.3–35 Hz, EOG: 0.3–30 Hz, EMG: 10–49 Hz), followed by z-score normalization. Each 30-second epoch is organized into sequences of 20 consecutive epochs for sleep staging.

**Cognitive** (Shin et al., 2018) provides simultaneous EEG, EOG, and NIRS recordings from 26 participants performing cognitive tasks. The N-back cognitive task is a working memory task with three difficulty levels (0-, 2-, 3-back), where participants decide whether each presented digit matches the one shown $n$ trials earlier. For preprocessing, EEG and EOG data are bandpass filtered at 0.5–50 Hz and resampled to 200 Hz across 28 channels. NIRS signals are lowpass filtered at 0.2 Hz and baseline-corrected using the $-5$ s to $-2$ s window.

### C.3. Baseline Methods

We provide detailed descriptions of baseline methods used in our experiments.

**General multimodal fusion methods** are applicable across all three tasks. LMF (Liu et al., 2018) performs efficient cross-modal integration through low-rank tensor decomposition. CTMWA (Zhang et al., 2024) employs crossmodal translation network with meta weight adaption for robust multimodal fusion. MMML (Wu et al., 2024) introduces multi-loss training strategies to jointly optimize modality-specific and fusion objectives. For these methods, we keep the fusion module intact and only adapt the encoder to support input modalities.

**Multimodal emotion recognition methods** for EAV dataset. MM-DFN (Hu et al., 2022) employs a graph-based dynamic fusion architecture that addresses information redundancy by modeling contextual relationships across multiple semantic representations to improve inter-modal complementarity. GA2MIF (Li et al., 2023) develops two-stage multi-source information fusion using Multi-head Directed Graph Attention networks (MDGATs) for contextual modeling and Multi-head Pairwise Cross-modal Attention networks (MPCATs) for cross-modal modeling. AGF-IB (Shou et al., 2024) introduces adversarial alignment and graph fusion via information bottleneck for cross-modal learning. Hyper-MML (Kang et al., 2026) proposes a hypergraph multi-modal learning framework that integrates EEG with audio and video information, featuring an Adaptive Brain Encoder with Mutual-cross Attention (ABEMA) module for EEG processing and an Adaptive Hypergraph Fusion Module (AHFM) to model higher-order relationships among multi-modal signals. CMERC (Tu et al., 2024) presents

a calibration framework that addresses prediction uncertainty in conversational emotion recognition through progressive curriculum learning, hybrid contrastive representation refinement, and confidence-based regularization. HEEGNet (Li et al., 2026) introduces hybrid Euclidean-hyperbolic neural networks for single modality EEG representation learning with improved cross-subject generalization. Among these methods, Hyper-MML and HEEGNet are the only methods designed to support EEG input. For the other methods, following (Kang et al., 2026), we replace text modality with EEG modality for fair comparison.

**Multimodal sleep staging methods** for ISRUC dataset. SalientSleepNet (SSNet) (Jia et al., 2021) employs a temporal fully convolutional network architecture with multi-scale extraction and multimodal attention mechanisms to identify salient wave patterns in sleep physiological signals. CrossModalSleepTransformer (CMST) (Mostafaei et al., 2024) employs transformer encoder-decoder architecture with cross-modality attention for integrating multiple physiological channels. CrossFusionSleepNet (CFSNet) (Cao et al., 2026) presents a parallel architecture that processes multimodal features from both temporal and spectral domains, utilizing cross-attention to model interdependencies between time and frequency representations. MMNet (Lin et al., 2023) develops multiview fusion networks with multiscale local feature extraction and cross-linked fusion for sleep analysis. XSleepFusion (Hu et al., 2025) proposes a dual-stage information bottleneck framework with evolutionary attention Transformer for interpretable multimodal sleep analysis.

**Hybrid EEG-fNIRS learning methods** for Cognitive dataset. STA-Net (Liu et al., 2025) proposes spatial-temporal alignment networks with functional near-infrared spectroscopy (fNIRS)-guided spatial alignment and EEG-guided temporal alignment to synchronize EEG and fNIRS signals. ST2A (Shi et al., 2025) introduces spatial-temporal convolution with dual attention mechanisms for EEG-fNIRS classification. EFDFNet (Xu et al., 2025) employs fNIRS features to enhance EEG feature disentanglement and uses a deep fusion strategy for effective multimodal feature integration, combining EMCNet (an attention state classification network for EEG that combines Mamba and Transformer) with fNIRS processing. TSMMF (Si et al., 2025) employs a bidirectional cross-modal transformer architecture for temporal-spatial fusion in affective BCI applications, building unified representations while maintaining separate modality branches to retain distinctive characteristics of EEG and fNIRS signals. EF-Net (Arif et al., 2024) is a CNN-based multimodal deep-learning model designed for subject-independent mental state recognition from EEG-fNIRS signals.

