# OpenReview forum: "EEG-Based Multimodal Learning via Hyperbolic Mixture-of-Curvature Experts"
_ICML.cc/2026/Conference — ICML 2026 regular_

### Official Review · Reviewer_UYY2 · 2026-03-03

**Soundness:** 4
**Presentation:** 4
**Significance:** 3
**Originality:** 4
**Overall Recommendation:** 5
**Confidence:** 3

**Summary:**

The Authors propose a multimodal data classification strategy, termed EEG-MoCE, that employs hyperbolic embeddings of individual inputs, attention-based fusion of embeddings with learnable individual curvatures, and Multinomial Logistic Regression classification in the hyperbolic space.
The core modality is EEG, and, depending on the application, additional data sources include facial video and audio recordings (for emotion assessment), EMG and EOG signals (for sleep staging), and EOG and infrared recordings (for cognitive assessment).
The Authors justify the selection of hyperbolic space for data analysis by the hierarchical structure of information provided by the EEG and the remaining data sources.
The proposed data analysis pipeline begins by deriving Euclidean-space embeddings of the inputs, computed independently for each modality. The resulting feature vectors are then projected onto hyperbolic manifolds with modality-specific, learned curvatures, and transformed via batch normalization, Lorentz activation, and pooling into hyperbolic embeddings.
Next, these embeddings are fused together to produce comprehensive feature vectors that aggregate information from all considered sources. To unify embeddings, they are projected to a unified fusion manifold prior to aggregation. The fusion is performed using the multi-headed attention scheme, where, for each attention layer, the head outputs are combined using the Frechet mean. Finally, the resulting embedding is used to perform a downstream task with a hyperbolic classifier.

**Compliance With Llm Reviewing Policy:**

Affirmed.

**Final Justification:**

The paper is sound, well-grounded in recent theoretical developments, and offers original ideas that have been positively verified through a series of well-designed experiments. The Authors demonstrate that hyperbollic spaces can be considered as an effective domain for tackling problems involving the analysis of hierarchically structured data, which makes the contribution significant for the considered domain. The paper is also easy to follow. Therefore, my original assessment of the paper was positive, and the rebuttal phase did not influence this opinion.

**Key Questions For Authors:**

Figure 1 could be improved by a more extensive explanation of the data distribution structures in Euclidean and hyperbolic spaces.

**Limitations:**

Yes

**Strengths And Weaknesses:**

The paper is sound, well-grounded in recent theoretical developments in hyperbolic neural networks. The Authors methodically and purposefully combine concepts, properties, and formal tools to build a systematic, convincing data-processing pipeline. They propose a curvature-guided cross-attention mechanism for aggregating information from different sources. Experimental verification of the method is extensive and based on adequate, difficult tasks. The provided ablation study clearly illustrates the roles of the algorithm's modules and justifies the proposed solutions. The provided results prove the potential of the proposed methodology, and the presented discussion addresses all adequate issues.

The paper introduces a few original contributions to the field. The main one is the original method for solving difficult data analysis problems involving information from multiple sources, where the contents of each modality have a complex, hierarchical structure. The Authors propose to employ an appropriate approach – hyperbolic neural networks, which are known to capture hierarchical relations much better than conventional Euclidean-space-based methods. The proposed data processing pipeline is very clear and well-motivated, and results in state-of-the-art performance. The second original contribution is a proposal to learn diverse hyperbolic-space curvatures for each modality, enabling better alignment of information content with the properties of the feature-extraction platform. Finally, the third novelty is the proposed curvature-guided cross-attention mechanism, appropriately adapted for hyperbolic computing, that enables optimal aggregation of relevant information across distinct modalities.

The problem considered by the Authors is difficult, and the proposed solution is a significant contribution to the field. First, the paper confirms the importance of using hyperbolic spaces as an adequate domain for both representation extraction and the execution of downstream machine learning tasks for hierarchically structured data. It also provides a template for performing multi-modal classification tasks in such spaces.

The paper has a logical, clear structure and is quite easy to follow. Also, it provides valuable appendices with rigorous formal definitions of all relevant concepts used in the presented data analyses.

---

> ### Author Rebuttal · Authors · 2026-03-30
>
> We greatly thank the reviewer for recognizing the contributions and novelty of our work!
> We also truly appreciate your careful reading of the manuscript and your constructive suggestion on Figure 1.
>
> ---
>
> **Figure 1: Euclidean vs. hyperbolic data distributions**
>
> Thank you for this helpful suggestion regarding Figure 1.
> We expanded the explanation of Figure 1 to help readers better understand why hyperbolic geometry is suitable for our setting.
>
> In the revised manuscript, we expanded both the caption and the accompanying explanation of Figure 1 to make the geometric contrast explicit.
> Euclidean space tends to under-represent hierarchical branching due to its flat geometry, whereas hyperbolic space better preserves tree-like separation through exponential volume growth.
> We also clarified how this geometric intuition connects to our multimodal scenario, where different modalities can exhibit different levels of hierarchy.
> Thank you again for this valuable suggestion, which helped us improve the clarity of the presentation.

---

> > ### Author Rebuttal · Reviewer_UYY2 · 2026-04-02
> >
> > My only concern was addressed in the provided response. I keep the original score (5 - Accept).

---

> > > ### Author Response · Authors · 2026-04-07
> > >
> > > Thank you again for your time and for your careful reading of our manuscript!
> > > We truly appreciate your feedback and support.

---

### Official Review · Reviewer_QmZi · 2026-03-12

**Soundness:** 3
**Presentation:** 3
**Significance:** 3
**Originality:** 3
**Overall Recommendation:** 5
**Confidence:** 4

**Summary:**

This paper presents a hyperbolic mixture-of-curvature experts framework (EEGMoCE) for multimodal neurotechnology. It leverages the hierarchical structures in EEG and associated modalities, which Euclidean embeddings struggle to represent. EEGMoCE assigns each modality to an expert in a learnable-curvature hyperbolic space and uses a curvature-aware fusion strategy to emphasize modalities with richer hierarchical information. The experimental results demonstrate the effectiveness of EEGMoCE.

**Compliance With Llm Reviewing Policy:**

Affirmed.

**Final Justification:**

My concerns have been adequately addressed. I raise my score to 5.

**Key Questions For Authors:**

See weaknesses

**Limitations:**

yes

**Strengths And Weaknesses:**

Strengths

1.The idea of using curvature as a proxy for hierarchical structure in fusion presented in this paper is original.

2.The theoretical derivations are rigorous, and all operations preserve manifold constraints.

3.The experiments are extensive and incorporate recent hyperbolic methods.

Weaknesses

1.The method description focuses too much on mathematical derivations, lacking an explanation of the actual role of the mathematical formulas within the model. A more effective approach would be to describe the formulas in the context of their specific application in the model. This would help readers unfamiliar with the field better understand the working principles and practical significance of the model.

2.The structure of the paper could be further streamlined and optimized. For example, the related work section is overly lengthy, which slows down the introduction to the main topic. It is recommended to introduce the core content more quickly, allowing readers to focus on the key aspects of the research sooner.

---

> ### Author Rebuttal · Authors · 2026-03-30
>
> We sincerely thank the reviewer for the positive feedback on our work! For your constructive suggestions on exposition and paper organization, we have revised the manuscript accordingly to improve clarity and readability.
>
> ---
>
> **1. Role of formulas in the model**
>
> Thank you for highlighting the need for more intuitive formula explanations.
> We agree that the role of several formulas could be explained more explicitly for readers less familiar with this area.
>
> In the revision, we retained the core formulas in the main text to preserve technical rigor, while adding targeted pipeline-level clarifications.
> A new appendix subsection provides equation-level notes on each formula's functional role in hyperbolic neural network computation.
> We also added a compact algorithm box that maps key equations to forward-pass steps, helping readers connect mathematical expressions with implementation logic.
>
> ---
>
> **2. Paper structure and related work**
>
> We appreciate this suggestion.
> In the revised manuscript, we have streamlined Section 2.1 by merging repetitive background and moving less essential literature details to the appendix.
> We have also moved the core method motivation earlier in the introduction so readers can reach the technical novelty more quickly.
> These changes improve readability and pacing while preserving clear positioning of our contributions.
>
> ---
>
> Thank you again for these helpful comments.
> They are very valuable for improving the final presentation quality of the paper.

---

> > ### Author Rebuttal · Reviewer_QmZi · 2026-04-02
> >
> > I fully understand and highly value the written explanation you provided, but I would like to see a clear response from you to my reply.

---

> > > ### Author Response · Authors · 2026-04-07
> > >
> > > We appreciate your acknowledgment of our explanations.
> > > Since the system does not permit a revised PDF upload at this stage, we have detailed our specific update comparisons below for each significant change.
> > >
> > > ---
> > >
> > > **1. Role of formulas**
> > >
> > > We have added a new 'Pipeline overview' paragraph to Section 3 to clarify the functional roles of core formulas:
> > >
> > > > **Pipeline overview.** From a pipeline perspective, the principal equations of EEG-MoCE define a sequential transformation: Equation (10) first embeds features into per-modality manifolds; Equation (8) then projects them onto a shared fusion manifold to enable cross-modal attention. Equation (6) and Equation (7) jointly parameterize the attention mechanism, using learned curvatures to control the weighting and sharpness of information integration. Following this, Equation (3) aggregates the attended features, and Equation (4) performs the final linear map in hyperbolic space before HMLR (Equation (27)). A detailed mapping of these equations to implementation steps and training updates is provided in Appendix A.4.
> > >
> > > We have included a new Appendix A.4 that explains the role of each equation within the hyperbolic neural network, alongside an algorithm pipeline box (available at https://anonymous.4open.science/r/EEG-MoCE/pic/Algo_box_Eq.png).
> > >
> > > The appendix text explains the equations referenced in the Sec. 3 pipeline paragraph above and adds the following.
> > > Regrettably, we can only summarize briefly here because of the character limit.
> > > It explains the scalar fusion curvature and details expert-internal operations, including gyrovector batch normalization (Eq. 20), Lorentz activation (Eq. 16), and Lorentz concatenation (Eq. 26).
> > > It also specifies the geodesic distance (Eq. 2) and the curvature prior in attention logits (Eq. 7), with a cross-reference for the HMLR classifier (Eq. 27).
> > >
> > > ---
> > >
> > > **2. Paper structure and related work**
> > >
> > > Our revised manuscript presents the method novelty earlier in the introduction, moves less essential literature to the appendix, and merges repetitive background.
> > >
> > > (i) We have replaced the paragraphs below in the introduction with a contribution list, making the novel aspects easier to locate.
> > >
> > > Submitted:
> > >
> > > > In this study, we propose an EEG hyperbolic mixture-of-curvature expert [...] in the fusion process.
> > >
> > > Revision:
> > >
> > > > In this work, we introduce EEG-MoCE, a hyperbolic mixture-of-curvature framework for EEG-based multimodal learning, with the following contributions:
> > > > - To our knowledge, the first systematic hierarchical analysis and hyperbolic framework for multimodal physiological signals.
> > > > - Per-modality experts with learnable curvatures that adapt to intrinsic modality differences.
> > > > - Curvature-guided fusion leveraging learned curvatures as proxies for hierarchical structure and modality importance.
> > > > - Extensive cross-subject experiments showing strong gains and SotA performance on three public EEG-based multimodal datasets.
> > >
> > > (ii) In Sec. 2.1, we have shortened the cognitive hierarchy discussion by moving the detailed examples of other modalities out of the main text and into a new Appendix C.1.
> > >
> > > Submitted:
> > >
> > > > Other modalities also exhibit hierarchical structures [...] hyperbolic geometry as a unified embedding framework.
> > >
> > > Revision:
> > >
> > > > Hierarchical structure is also implicit in the mechanisms of other modalities (Appendix C.1). Accordingly, hyperbolic geometry supports embeddings of hierarchical data, motivating a unified geometric view of all modalities in EEG-based multimodal learning.
> > >
> > > In the main text, we condensed the EEG-based multimodal learning paragraph into a single sentence by moving the list of Euclidean baselines to the 'Baseline methods' subsection in the appendix.
> > >
> > > Submitted:
> > >
> > > > To address these challenges, various architectures [...] extending hyperbolic multimodal methods to EEG-based multimodal learning.
> > >
> > > Revision:
> > >
> > > > To address these challenges, various Euclidean architectures have been proposed, as detailed in Appendix Baseline methods, whereas hyperbolic multimodal learning methods have not yet been explored for EEG-based multimodal learning.
> > >
> > > (iii) We merged the previous related work blocks 'Hyperbolic neural networks' and 'Mixed-curvature learning' into a single block to reduce redundancy.
> > >
> > > Submitted:
> > >
> > > > These methods focus on single-modality EEG with fixed curvature, [...], varying degrees of hierarchical structure as demonstrated empirically in Section 4.1.
> > >
> > > Revision:
> > >
> > > > These methods for neuroscience applications focus on single-modality EEG with fixed curvature. Recent work has shown that for a single textual modality, different components or tasks may exhibit varying degrees of hierarchical structure that are better captured by mixture-of-curvature models. [...] In this work, we generalize mixture-of-curvature modeling to EEG-based multimodal learning, given that different modalities may exhibit varying degrees of hierarchical structure.
> > >
> > > We hope the excerpts provided above make the revisions concrete.

---

### Official Review · Reviewer_kK88 · 2026-03-17

**Soundness:** 2
**Presentation:** 3
**Significance:** 2
**Originality:** 2
**Overall Recommendation:** 3
**Confidence:** 4

**Summary:**

This paper is about EEG-MoCE, a multmodal framework fro EEG based neorotechnology that embeds each modality in its own learnable curvature hypoboloc space and perfroems curvature aware cross modal fusion. It is evaluated on three multimodal datasets covering different tasks.

**Compliance With Llm Reviewing Policy:**

Affirmed.

**Key Questions For Authors:**

1. Computational cost analysis
2. Interpretation of curvature
Eq 7 explicitly biases the attention towards larger curvature modalities. How should the observation that high curvature modalities contribute more to the fusion be interpreted as evidence of modality importance rather than a consequence of the equation?

3. Fusion design
Why is the arithmetic mean the appropriate choice? Would a learnable fusion curvature oor product manifold be better?

4. Robustness
Given the small size of the dataset. could the paper report multi seed or boostrap analysis for stability in reported improvements?

5. Role of hierarchy
Clarify how hyperbolicity metic directly relates to downstream task performance?

**Limitations:**

Partially. THe authors metion computational stability of the hyperbolic operation but do not fully discuss the computational overheads, dataset size, heuritic nature, etc. A more detailed limitation would strengthen the paper.

**Strengths And Weaknesses:**

Strengths

1. Integration of geometric learning and neurotechnology.
Paper combines several ideas from geometric deep learning and adapts them to neurotechnology
2. Evaluation across multiple tasks
3. Ablation studies
4. Clear presentation

Weakness

1. Limited Novelty
The core elements presented in the paper are prior works. The only novelty is adaption of prior work for EEG-based learning rather than introducing a new framework

2. Circular interpretation
The curvature fusion mechanishm biases attention towards modalities with larger curvature. Later the paper reports that modalities with higher vcurevature contribute more than fused representation. The architecture itself enforces this bias, this observation risks partially a consequence of the model design rather than a discovery about modality importance.

3. Fusion across manifolds
The paper acknowledge that fusing from manifolds with different curvatures is ill defined.

4. Lack of computational cost
Hyperbolic neural networks are computationally expensive. The papers domain is brain-computer interfaces where latency matters, the paper would benefit ffrom reporting computational costs

5. Statistical robustness
While the paper reprots mean+- std dev across folds, prior works have shown that benchmark improvements can fall within evaluation variance when examined under multiple seeds (Henderson et al 2018, Dodge et al 2019) The paper would benefit from additional analysis such as multi seed experiments, or bootstrap CI.

6. Limited justification for heirarchial structure
The paper relies heavily on assumption that the modalities exhibit hierarchial structure that benifit from hyperbolic. The hyperbolicity provices some evidence, but does not demonstrate that it is the primary driver.

---

> ### Author Rebuttal · Authors · 2026-03-30
>
> Thank you for your careful review.
> We have grouped related weaknesses (W) and questions (Q), and we hope it meets your high standards.
>
> ---
>
> **1. Novelty of our framework (W1)**
>
> Previous work has focused on constant curvature [1] and single-modality settings [3].
> As acknowledged by reviewer UYY2, we first extended [1] to the multimodal setting; we then proposed per-modality learnable curvatures that adapt to intrinsic modality differences; and we proposed curvature-guided fusion using learned geometric structure.
>
> ---
>
> **2. Interpretation of curvatures and modality contributions (W2, Q2)**
>
> Actually, Table 2 is run **without** Eq. 7 (no curvature bias), while modality curvatures remain learnable.
> Modalities with larger learned curvature still exhibit a greater contribution, which motivates our curvature-guided fusion design.
>
> Table 3 then adds Eq. 7 and shows that $\lambda$ consistently increases from its initialization across datasets, indicating that the model learns to rely on curvature information for attention weighting. This further supports the effectiveness of our design.
>
> ---
>
> **3. Fusion design across manifolds (W3, Q3)**
>
> Following [3,4], we project each modality into a shared fusion manifold with curvature-dependent scaling (Eq. 8), which is well-defined between Lorentz models.
>
> The manuscript uses an elegant arithmetic mean for the fusion curvature.
> We also tried a learnable fusion curvature but prefer the mean, since the learnable variant performs comparably or slightly worse:
>
> | Dataset | Fusion | Acc (%) | F1 (%) |
> | --- | --- | --- | --- |
> | EAV | Mean | 75.88 ± 8.31 | 75.47 ± 8.66 |
> | | Learnable | 75.55 ± 8.57 | 75.10 ± 9.13 |
> | ISRUC | Mean | 78.53 ± 2.95 | 75.38 ± 4.05 |
> | | Learnable | 78.08 ± 3.01 | 74.97 ± 3.96 |
> | Cognitive | Mean | 62.39 ± 13.07 | 59.67 ± 13.08 |
> | | Learnable | 60.96 ± 13.48 | 57.68 ± 13.61 |
>
> ---
>
> **4. Computational cost and latency (W4, Q1)**
>
> We report time cost on EAV for ours and the strongest Euclidean baselines, all measured on the same RTX 4090 GPU.
>
> | Method | Train sec/epoch | Test msec/trial |
> | --- | ---: | ---: |
> | MMML | 10.86 | 1.17 |
> | Hyper-MML | 12.95 | 1.24 |
> | GA2MIF | 8.56 | 0.86 |
> | Ours | 26.07 | 2.71 |
>
> As expected, hyperbolic operations add more computational cost.
> However, given the high performance we achieved, this is acceptable.
> On EAV, inference takes only 2.71 ms per 20 s trial, which remains negligible for online BCI use.
>
> ---
>
> **5. Statistical robustness (W5, Q4)**
>
> The variance reported in the main paper is computed across subjects, and we already use the largest publicly available multimodal EEG benchmarks we could identify (e.g., EAV with 42 subjects).
>
> While [5] discusses caveats of k-fold significance testing in RL, it acknowledges that such testing remains standard in supervised settings like ours.
> We nonetheless followed your recommendation with a multi-seed robustness check.
> We report seed-level statistics aggregated across 5 random seeds.
> Seed-level variation is smaller than subject-level variation, and the observed improvements remain significant across seeds.
>
> | Dataset | Method | Acc (%) | F1 (%) |
> | --- | --- | --- | --- |
> | EAV | Ours | 75.79 ± 0.78 | 75.25 ± 0.76 |
> | | HEEGNet | 61.54 ± 0.76 | 60.37 ± 0.77 |
> | ISRUC | Ours | 78.42 ± 0.31 | 75.10 ± 0.29 |
> | | XSleepFusion | 75.02 ± 0.67 | 73.30 ± 0.71 |
> | Cognitive | Ours | 61.86 ± 1.75 | 59.23 ± 1.64 |
> | | EF-Net | 54.28 ± 2.41 | 51.36 ± 2.35 |
>
> Due to the limited rebuttal time, multi-seed runs for additional baselines are in progress.
>
> We also computed subject-level bootstrap 95% CIs by resampling subjects 2,000 times.
> For EAV, the CIs are [0.734, 0.783] for Accuracy and [0.729, 0.780] for F1.
>
> ---
>
> **6. Hierarchical structure and performance (W6, Q5)**
>
> We motivate hyperbolic geometry with hierarchical structure as an inductive prior, consistent with prior hyperbolic representation learning in EEG [1], vision [2], and language [3].
>
> Admittedly, there may not be a strict monotonic relationship between a hyperbolicity metric and absolute downstream gains across tasks.
> Performance improvements are jointly affected by many factors (task difficulty, data protocols, etc.), so gain magnitudes are not directly comparable across datasets.
> This is also consistent with empirical evidence in prior work (Tables 1 and 2 in [1]).
>
> On hyperbolic performance benefits, Table 7 shows clear drops when hyperbolic components are removed; Figure 3 (t-SNE) further suggests improved class separability in the learned hyperbolic embeddings.
>
> ---
>
> [1] Li et al. *HEEGNet: Hyperbolic Embeddings for EEG*. ICLR, 2026.
>
> [2] Bdeir et al. *Fully Hyperbolic Convolutional Neural Networks for Computer Vision*. ICLR, 2024.
>
> [3] He et al. *HELM: Hyperbolic Large Language Models via Mixture-of-Curvature Experts*. NeurIPS, 2025.
>
> [4] Yang et al. *Hypformer: Exploring Efficient Transformer Fully in Hyperbolic Space*. KDD, 2024.
>
> [5] Henderson et al. *Deep Reinforcement Learning that Matters*. AAAI, 2018.

---

> > ### Author Rebuttal · Reviewer_kK88 · 2026-04-02
> >
> > 1.  Correlation between curvature bias and modality is partially resolved.
> > 2. The reported seed level variance being smaller than subject-level confidence that gains are not due to initialization noise. Resolved
> > 3. Computational cost is helpful
> >
> >
> > Main concerns remain
> > 1. Novelty is incremental -- It is largely an extension of prior work of existing hyperbolic and mixed curvatures,
> > 2. Fusion is heuristic -- The justificaltion for use of mean is emperical rather than principled. The rebuttal shows that a learnable curvature performs slightly worse, this does not fully resolve my concern that projecting into single average may distort modality specific geometry.

---

> > > ### Author Response · Authors · 2026-04-07
> > >
> > > Thank you for the acknowledgment of our previous rebuttal and for clearly stating the two remaining concerns.
> > >
> > > ---
> > >
> > > **1. Regarding novelty**
> > >
> > > Our work is a neuroscience application track paper that aims to advance the neuroscience domain.
> > > It presents the first systematic hierarchical analysis for multimodal decoding of physiological signals and proposes the first hyperbolic multimodal framework, thereby opening a new research direction for this application domain.
> > >
> > > We provide the first intrinsic hierarchical analysis for multimodal EEG data to support future work.
> > > Table 1 shows a tree-like structure across modalities, with modality-dependent strength of that structure.
> > > Tables 2 and 3 are consistent with the insight that learned curvature reflects geometric structure and informational richness.
> > >
> > > Our methodological contributions are closely aligned with the analysis above and are not superficial add-ons to prior mixture-of-curvature work such as [3].
> > >
> > > 1. Per-modality experts.
> > > Differences in hierarchical structures across modalities motivate per-modality hyperbolic experts that adapt to intrinsic modality differences.
> > > This contribution follows the nature of multimodal physiological data.
> > >
> > > 2. Curvature-guided fusion.
> > > Building on Tables 1-3 and modality experts, we route fusion using a curvature-informed structure, including curvature-guided temperature and prior.
> > > This design is absent from prior hyperbolic work and is central to our multimodal pipeline.
> > >
> > > Other reviewers have also acknowledged these aspects of novelty.
> > > The performance gains from our framework are promising.
> > > Thorough experiments and ablations show that performance does not arise from reusing a hyperbolic encoder on its own.
> > > All our novelties for multimodal EEG are effective.
> > >
> > > ---
> > >
> > > **2. Fusion curvature concern**
> > >
> > > The projection onto a unified manifold is a pragmatic strategy; several works [3,7,8] implement a similar strategy by mapping onto a unified manifold.
> > > [7] and [8] use gated mapping with fusion and outcomes in Euclidean space, while [3] routes multi-curvature branches and projects them onto a unified hyperbolic backbone.
> > > A product manifold gives straightforward concatenation [6], but lacks cross-modal interaction support.
> > >
> > > We set $K_{\mathrm{fus}}$ to the mean of expert curvatures also because it minimizes $\sum_m(K_{\mathrm{fus}}-K_m)^2$ in closed form, keeping $K_{\mathrm{fus}}$ near each $K_m$.
> > > Below, we relate this choice to geodesic distortion and to alternative curvature designs.
> > >
> > > (i) [4] introduced the widely adopted curvature-scaling mechanism we used in Eq. 8, which preserves the ordering of pairwise Lorentz geodesic distances.
> > > After this, larger curvature gaps imply stronger global rescaling of geodesic distances on the new manifold.
> > > We therefore prioritize a fusion scalar that stays simultaneously close to all $\{K_m\}$, which we take to be the mean.
> > >
> > > (ii) [9] is concurrent work from ICLR 2026 and the first to study learning an intermediate manifold curvature for hyperbolic manifold alignment.
> > > They approximate the KL divergence between wrapped normals to define a manifold distance $D_L$ and then optimize a bridge curvature between two manifolds.
> > > This yields a principled estimate of the bridge curvature; we therefore use it as a reference for comparison.
> > >
> > > Their objective $D_L$ is coupled to the gap between each modality curvature and the bridge curvature, which is consistent in spirit with our mean fusion choice.
> > > However, their minimization requires additional iterations and hyperparameters, which increases instability.
> > >
> > > We ran the same comparison on our benchmarks with pretrained per-modality curvatures held fixed:
> > >
> > > | Dataset | Mean $K$ | Opt $K$ |
> > > |--------|------------------|-------------------------------------|
> > > | EAV | -2.2544 ± 0.1020 | -2.2509 ± 0.1013 |
> > > | ISRUC | -1.3511 ± 0.0777 | -1.3547 ± 0.0760 |
> > > | Cognitive | -1.9834 ± 0.0630 | -1.9821 ± 0.0623 |
> > >
> > > Mean fusion curvature statistically matches the curvature obtained by minimizing the objective in [9].
> > > TOST on paired $K$ means with margin $\pm 0.005$ supports equivalence on all three benchmarks and is even smaller than typical variation from hyperparameter choices in [9].
> > > Similarly, in terms of performance, there is no clear difference in accuracy and F1 across datasets.
> > >
> > > We aim to provide a stable baseline for follow-up work in this direction.
> > > Therefore, the mean is reasonable and especially appropriate for simplicity and stability.
> > > Admittedly, the Limitations section of the revision also notes richer fusion and transport mechanisms, including objectives in the spirit of [9], as future work.
> > >
> > > ---
> > >
> > > [6] Chen and Sui. *PROCLIP: Product Space Multimodal Contrastive Alignment*. GRaM at ICLR 2026.
> > >
> > > [7] Gao et al. *Mixed-Curvature Multi-Modal Knowledge Graph Completion*. AAAI, 2025.
> > >
> > > [8] Chen et al. *GalaxyWalker: Geometry-aware VLMs For Galaxy-scale Understanding*. CVPR, 2025.
> > >
> > > [9] Wu et al. *Modality Alignment Across Trees on Heterogeneous Hyperbolic Manifolds*. ICLR, 2026.

---

### Decision · Program_Chairs · 2026-04-30

**Decision:**

Accept (regular)

**Comment:**

The paper received 2 Accepts and one weak reject. Overall, the reviewers are generally positive on the work.

One reviewer noted limited novelty, however, others noted the idea is original and the contribution is significant.

Other strengths include, but are not limited to, rigorous theoretical derivations, strong experiments and ablations across multiple tasks, and a clear presentation.

The authors submitted a detailed rebuttal responding to reviewer concerns such as limited novelty, and missing justifications. The work is interesting, and the authors have done a good job responding to reviewer concerns.